# Stein Variational Goal Generation for adaptive Exploration in Multi-Goal Reinforcement Learning

## Abstract

Multi-goal Reinforcement Learning has recently attracted a large amount of research interest. By allowing experience to be shared between related training tasks, this setting favors generalization for new tasks at test time. However, in settings with discontinuities in the goal space (e.g. walls in a maze) and when the reward is sparse, a majority of goals are difficult to reach. In this context, some curriculum over goals is needed to help agents learn by adapting training tasks to their current capabilities. In this work we propose a novel approach: Stein Variational Goal Generation (SVGG), which builds on recent automatic curriculum learning techniques for goal-conditioned policies. SVGG samples goals of intermediate difficulty for the agent, by leveraging a learned predictive model of its goal reaching capabilities. In that aim, it models the distribution of goals with particles and relies on Stein Variational Gradient Descent to dynamically attract the goal sampling distribution in areas of appropriate difficulty. We show that SVGG outperforms state-of-the-art multi-goal Reinforcement Learning methods in terms of success coverage in hard exploration problems, and demonstrate that our approach is endowed with a useful recovery property when the environment changes.

## 1 Introduction

In standard Reinforcement Learning (RL), agents learn a policy to optimally achieve a single task. By contrast, in Multi-goal RL (Kaelbling, 1993), they address a set of tasks by having policies conditioned on goals, where each goal corresponds to an individual task. The resulting goal-conditioned policies offer an efficient way for sharing experience between related tasks (Schaul et al., 2015; Pitis et al., 2020b; Yang et al., 2021). In general, the corresponding agents are equipped with a capability to sample goals from some goal space, but do not know whether a given goal can be achieved or not. Besides, the general ambition in the multi-goal RL context is to obtain an agent able to reach any goal from a desired goal distribution and to do so reliably and efficiently. This is particularly challenging in settings where the desired goal distribution is unknown at train time, which means discovering the goal space by experience and optimizing its coverage without any prior knowledge.

To avoid deceptive gradient issues, multi-goal RL often considers the sparse reward context, where the agent only obtains a non-null learning signal when the goal is reached. In that case, the multi-goal framework makes it possible to leverage Hindsight Experience Replay (HER) (Andrychowicz et al., 2017) which helps densify the reward signal by relabeling failures as successes for the goals achieved by accident. However, in settings with discontinuities in the goal space (e.g., walls in a maze), or in hard-exploration problems where the long task horizon results in an exponential decrease of the learning signal (Osband et al., 2016), many goals remain hard to reach.

In these more difficult contexts, and without any desired goal distribution at hand, we want to maximize the performance of the agent on all feasible goals, that we call the success coverage. This metric encompasses the capacity of the agent to explore and master every goal in the environment. To do so, the selection of goals must be structured into a curriculum to help agents to explore and learn progressively by adapting training tasks to their current capabilities (Colas et al., 2018). The question is: how can we organize a curriculum of goals that maximize the success coverage?

A first approach consists in focusing on novelty, with the objective of expanding the set of achieved goals Pitis et al. (2020b), Pong et al. (2019), Warde-Farley et al. (2018), Nair et al. (2018). This leads to strong exploration results, but success coverage is only optimized implicitly. Another strategy is to bias the goal generation process toward goals of intermediate difficulty (GOIDs) that will intuitively provide a strong learning signal to the agent Florensa et al. (2017), Racaniere et al. (2019), Sukhbaatar et al. (2017), Zhang et al. (2020). By aiming at performance, those methods target more explicitly success in encountered goals, but benefit from implicit exploration.

In this work, we propose a novel method which provides the best of both worlds. Our method, called SVGG[1], learns a model of the probability of succeeding in reaching goals, and targets goals whose success is the most unpredictable. To model such a distribution over the goal space, we rely on a set of particles where each particle represents a goal candidate. This set of particles is updated via Stein Variational Gradient Descent (Liu & Wang, 2016) to fit our objective of goals of intermediate difficulty. Due to the optimization properties of SVGD, in the absence of goals of intermediate difficulty, the current particles will repel one another and foster exploration. We use this feature to demonstrate that SVGG possesses a very useful *recovery property* that prevent from catastrophic forgetting and enables the agent to adapt when the environment changes during training.

## 2 Background and Related Work

In this paper, we consider the multi-goal reinforcement learning setting, defined as a Markov Decision Process (MDP) $\mathcal{M}_g = <S, T, A, g, R_g>$, where $S$ is a set of states, $T$ is the set of transitions, $A$ the set of actions and the reward function $R_g$ is parametrized by a goal $g$ lying in the d-dimensional continuous goal space $\mathcal{G} \equiv \mathbb{R}^d$. In our setting, each goal $g$ is defined as a set of states $S_g \subseteq S$ that are desirable situations for the corresponding task, with states in $S_g$ being terminal states of the corresponding MDP. Thus, a goal $g$ is considered achieved when the agent reached at step $t$ any state $s_t \in S_g$, which implies the following sparse reward function $R_g : S \to \{0; 1\}$ in the absence of expert knowledge. With $I$ the indicator function, $R_g(s_t, a_t, s_{t+1}) = I(s_{t+1} \in S_g)$ for discrete state spaces and $R_g(s_t, a_t, s_{t+1}) = I(\min_{s^* \in (S_g)} ||s_{t+1} - s^*||_2 < \delta)$ for discrete ones, where $\delta$ is a distance threshold.

Then, the objective is to learn a goal-conditioned policy (GCP) $\pi : S \times \mathcal{G} \to A$ which maximizes the expected cumulative reward from any initial state of the environment, given a goal $g \in \mathcal{G}$: $\pi^* = \arg\max_\pi \mathbb{E}_{g \sim p_g} \mathbb{E}_{\tau \sim \pi(\tau)} [\sum_{t=0}^{\infty} \gamma^t r_t^g]$, where $r_t^g = R_g(s_t, a_t, s_{t+1})$ stands for the goal-conditioned reward obtained at step $t$ of trajectory $\tau$ using goal $g$, $\gamma$ is a discount factor in $]0; 1[$ and $p_g$ is the distribution of goals over $\mathcal{G}$. In our setting we consider that $p_g$ is uniform over $S$ (i.e., no known desired distribution), while the work could be extended to cover different distributions.

### 2.1 Automatic Curriculum for sparse Reward RL

Our SVGG method addresses automatic curriculum for sparse reward goal-conditioned RL (GCRL) problems and learns to achieve a continuum of related tasks.

**Achieved Goals Distributions** Our work is strongly related to the MEGA algorithm (Pitis et al., 2020b), which (1) maintains a buffer of previously achieved goals, (2) models the distribution of achieved goals via a kernel density estimation (KDE), and (3) uses this distribution to define its behavior distribution. By preferably sampling from the buffer goals at the boundary of the set of already reached states, an increase of the support of that distribution is expected. Doing so, MEGA aims at overcoming the limitations of previous related approaches which also model the distribution of achieved goals. For instance, DISCERN (Warde-Farley et al., 2018) only uses a replay buffer of goals whereas RIG (Nair et al., 2018) and Skew-fit (Pong et al., 2019) rather use variational auto-encoding (Kingma & Welling, 2013) of the distribution. While RIG samples from the modeled achieved distribution, and DISCERN and Skew-Fit skew that distribution to sample more diverse achieved goals, MEGA rather focuses on low density regions of the distribution, aiming to expand it. This results in improved exploration compared to competitors. Our work differs from all these works as they only model achieved goals, independently from which goal was targeted when they were

---

[1]For Stein Variational Goal Generation. Code and instructions to reproduce our results will be made publicly available at https://anonymous.4open.science/r/Stein-Variational-Goal-Generation-1A6E.

achieved, whereas we model the capability of reaching target goals. This makes a strong difference since, while MEGA selects goals at the frontier of what it already discovered, nothing indicates that goals $g$ closer to the mode of the distribution can be achieved when they are targeted. MEGA is also prone to catastrophic forgetting and limits exploration to goals present in the replay buffer.

**Adversarial Goal Generation**    Another trend proposes to adversarially learn a goal generator, that produces targets that are at the frontier of the agent's capabilities. In that vein, GoalGAN (Florensa et al., 2018) simultaneously learns a discriminator to sort out non GOIDs or generated goals from GOIDs in the buffer of achieved goals, and a generator that aims at producing goals that fool the discriminator. While appealing, the approach is prone to instabilities, with a generator that may usually diverge far from the space of reachable goals. Setter Solver (Racaniere et al., 2019) stands as an extension of GoalGAN, where a goal setter is learned to provide goals of various levels of difficulty w.r.t. a judge network (similar to our success predictor, see next section). The provided goals remain close to already achieved goals, and are diverse enough to avoid mode collapse. This approach suffers from relying on invertible networks to map from the latent space to the goal space, which severely limits the modeling power, and can reveal problematic for high dimensional goal spaces, e.g. images. Asymmetric self-play (Sukhbaatar et al., 2017) is another way to generate goals, with a teacher agent seeking to produce goals that are just beyond the capabilities of the student agent. Both teacher and student learn simultaneously, with an equilibrium of adverse rewards determined on their respective time to go. However, this balance is hard to maintain, and many useful areas are missed. Finally, curiosity-driven exploration (Pathak et al., 2017) could be placed in that adversarial category, with an inverse model that aims at learning a compact latent representation space from which it is still possible to predict the dynamics of the environment. The agent is then rewarded for exploring areas whose dynamics are hard to predict. Nevertheless, it requires complex model learning and may lead to hallucinations for the agent which can remain stuck in difficult areas of the search space, by contrast with our simpler model-free setting.

**Skills Discovery**    Many recent works propose to deal with the problem of exploration based on a quantization of the state space in distinct skills, that are supposed to correspond to homogeneous sets of states that can be achieved with similar policies. Based on variational empowerment (Gregor et al., 2016; Choi et al., 2021) that approximates a posterior skill distribution via variational inference and mutual information maximization, DIAYN (Eysenbach et al., 2018) leverages a discriminator network to identify for each trajectory the skill that the agent is currently using. Extensions such as Kamienny et al. (2021) propose to sequentially chain skills to deal with hard environments. However, these approaches are unstable, and the full exploration of the environment is not guaranteed.

**Population-based Exploration**    Population-based RL, using evolutionary strategies to explore environments, is receiving increasing attention for hard exploration problems. Methods such as Novelty Search or Quality Diversity (Conti et al., 2018; Liu et al., 2018; Paolo et al., 2021) deal with populations of agents that evolve to maximize diversity objectives. However, such approaches are usually greatly less sample efficient than GCRL ones. Our SVGG approach can be considered as a trade-off between classical GCRL and evolutionary methods, by dealing with a population of goal particles, ensuring diversity and benefiting from sample efficiency of gradient-based optimization.

## 2.2 STEIN VARIATIONAL GRADIENT DESCENT

Our method builds on Stein Variational Gradient Descent (SVGD) (Liu & Wang, 2016) to approximate the distribution of goals of interest. SVGD is a powerful non-parametric tool for density estimation, when the partition function of the target distribution $p$ to approximate is intractable. It stands as an efficient alternative to MCMC methods, which are proven to converge to the true distribution $p$ but are usually too slow to be used in complex optimization processes. It also stands as an alternative to variational inference of parametric neural distributions $q$, which are restricted to pre-specified families of distributions (e.g., Gaussian or mixtures of Gaussians) that may not fit target distributions. Instead, it models $q$ as a set of particles $\{x_i\}_{i=1}^n$, all belonging to the support of $p$.

The idea behind SVGD is to approximate the target distribution $p$ with $q$ by minimizing their KL-divergence: $\min_q KL(q||p)$. This objective is reached by iterative deterministic transforms as small perturbations of the identity map, on the set of particles: $T(x) = x + \epsilon\phi(x)$, where $\phi$ is a smooth transform function that indicates the direction of the perturbation, while $\epsilon$ is the magnitude.

The authors draw a connection between KL-divergence and the *Stein operator* by showing that $\nabla_\epsilon KL(q_{[T]}||p)|_{\epsilon=0} = -\mathbb{E}_{x \sim q}[\text{trace}(\mathcal{A}_p \phi(x)]$, where $q_{[T]}$ is the distribution of particles after the transformation $T$, $\mathcal{A}_p \phi(x) = \phi(x)\nabla_x \log p(x)^T + \nabla_x \phi(x)$ being the Stein operator. The KL minimization objective is thus directly linked to the *Stein Discrepancy*, defined as: $\mathbb{S}(q,p) = \max_{\phi \in \mathcal{F}} \mathbb{E}_{x \sim q}[\text{trace}(\mathcal{A}_p \phi(x)]$. Note that $\mathbb{S}(q,p) = 0$ only if $q = p$.

Minimizing Stein Discrepancy being intractable as such, Liu et al. (2016) and Chwialkowski et al. (2016) introduce the *Kernelised Stein Discrepancy* (KSD) where the idea is to restrict to projections $\phi$ that belong to the unit ball of a reproducing kernel Hilbert space $\mathcal{H}$ (RKHS), for which there is a closed form solution. The KSD is defined as $\mathbb{S}(q,p) = \max_{\phi \in \mathcal{H}}\{\mathbb{E}_{x \sim q}[\text{trace}(\mathcal{A}_p \phi(x)], \quad s.t \quad ||\phi||_\mathcal{H} \leq 1\}$, whose solution is given by: $\phi^*(.) = \mathbb{E}_{x \sim q}[\mathcal{A}_p k(x,.)]$, where $k(x,x')$ is the positive definite kernel of the RKHS $\mathcal{H}$. The RBF kernel $k(x,x') = \exp(-\frac{1}{h}||x-x'||_2^2)$ is commonly used.

Therefore, the steepest descent on the KL-objective is given by the optimal transform:

$$x_i \leftarrow x_i + \epsilon \phi^*(x_i), \quad \forall i = 1 \cdots n, \text{where}$$

$$\phi^*(x_i) = \frac{1}{n}\sum_{j=1}^n \big[ \underbrace{k(x_j,x_i)\nabla_{x_j} \log p(x_j)}_{\text{attractive force}} + \underbrace{\nabla_{x_j} k(x_j,x_i)}_{\text{repulsive force}} \big]. \tag{1}$$

The "attractive force" in the update 1 drives the particles toward high density areas of the target $p$. The "repulsive force" pushes the particles away from each other, therefore fosters exploration and avoids mode collapse. Note that if $n = 1$, the update in (1) corresponds to a Maximum a Posteriori.

SVGD has already been successfully explored in the context of RL. The Stein Variational Policy method Gradient (SVPG) (Liu et al., 2017) employs SVGD to maintain a distribution of useful agents as particles. It strongly differs from our approach, since we consider particles as behavior goal candidates, while SVPG aims at capturing the epistemic uncertainty about policy parameters. A mix between this orthogonal work and SVGG could be considered as future work, to deal with policy and models uncertainty in addition to a useful distribution of GOIDs. Chen et al. (2021) also relies on SVGD to build a strategy to generate goals to agents, but in a very simplified setting without the attractive force from (1), which does not allow to fully benefit from this theoretical framework. Notably, such a kind of approach is particularly sensitive to catastrophic forgetting.

## 3 STEIN VARIATIONAL GOAL GENERATION

In this section we introduce our Stein Variational Goal Generation (SVGG) algorithm. Our aim is to obtain a curriculum to sample goals of appropriate difficulty for the RL agent, where the curriculum is represented as an evolving goal sampling probability $p_{\text{goals}}$. To do so, we maintain a model of the agent's skills – or goal reaching capability – and sample goals with the highest entropy, resulting in a distribution $p_{\text{skills}}$ which assigns probability mass to areas of goals of appropriate difficulty, relying on a model of the agent's current skills. Additionally, with a simple one class SVM, we learn a prior probability $p_{\text{valid}}$ preventing the particles from moving too far away from the previously achieved goals, and thus from ending up in non-feasible areas of the goal space. Thus we sample goals from the following target distribution:

$$p_{\text{goals}}(g) \propto p_{\text{skills}}(g).p_{\text{valid}}(g). \tag{2}$$

Formal definitions of these two components are given below.

**Model of the agent's skills** The probability $p_{\text{skills}}$ is modelled as a Neural Network $D_\phi$ whose parameters $\phi$ are learned by gradient descent on the following Binary Cross Entropy (BCE) loss:

$$\mathcal{L}_\phi = \sum_{(g^i,s^i) \in O} s^i(\log D_\phi(g^i)) + (1-s^i)(\log(1-D_\phi(g^i))), \tag{3}$$

where $O = \{g^i, s^i\}_{i=1}^{n_B}$ is a batch of $(goal, success)$ pairs coming from recent trajectories of the agent in the environment. The sampled goals are those whose predicted probability of success is neither too high nor too low (i.e., we avoid $D_\phi(g) \approx 1$ or $D_\phi(g) \approx 0$).

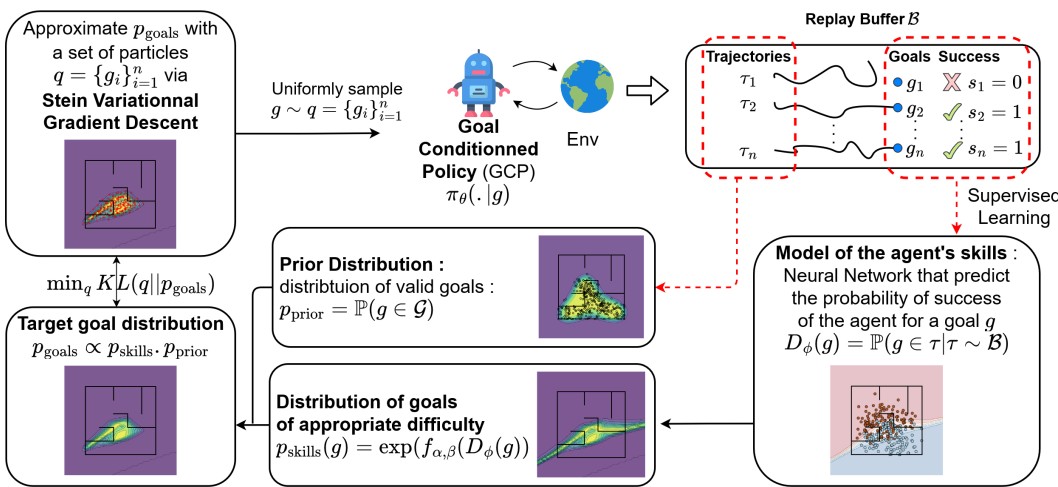

Figure 1: Overview of the SVGG method. The interaction of agents with their environment is stored in a replay buffer (top right) and used to learn $D_\phi$, a model of its abilities to reach goals (bottom right). We build on this model to compute a distribution of goals of appropriate difficulty $p_{\text{skills}}$, leveraging a prior $p_{\text{prior}}$ to stay inside the valid goal space. The obtained target goal distribution $p_{\text{goals}}$ is approximated with a set of particles $\{g_i\}_{i=1}^n$ using Stein Variational Gradient Descent (SVGD).

To build $p_{\text{skills}}$ based on the prediction of $D_\phi$, we use a beta distribution whose maximum density point mass is determined according to the output of $D_\phi$, by two hyper-parameters $\alpha$ and $\beta$ that shape the distribution, as illustrated in Figure 2. We report results for these 5 depicted settings in Appendix C.2 (unless specified otherwise, the setting we use is the *Medium* one from the figure).

We define the distribution $p_{\text{skills}}$ as an energy-based density whose potential is the output of the beta distribution $f$:

$$p_{\text{skills}}(g) \propto \exp\left(f(\alpha, \beta, D_\phi(g)\right). \tag{4}$$

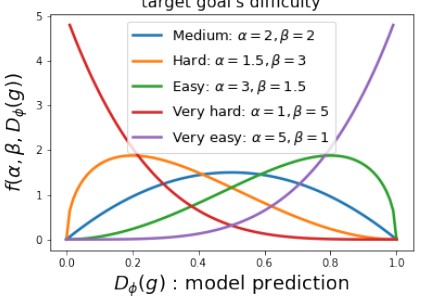

Figure 2: Modulation of goal's target difficulty with beta distributions.

**Prior distribution** To keep the sampled goals in the valid goal space of the environment $\mathcal{G}$, we need to define a prior distribution over the set of particles. We assume that the set of valid goals $\mathcal{G}$ is not given. However, the set of states already reached by the agent are stored in an archive $\mathcal{R}$.

We would ideally have a prior that considers the posterior probability that a goal $g$ belongs to $\mathcal{G}$ given $\mathcal{R}$: $p_{\text{valid}}(g) \propto \mathbb{P}(g \in \mathcal{G}|\mathcal{R})$. We progressively build such a distribution with a One Class SVM (OCSVM). This model is mainly designed for outlier or novelty detection in absence of labeled data. Given a dataset $X \in \mathbb{R}^d$, it defines a boundary of the data support in $\mathbb{R}^d$, while keeping a small portion of the data points out of that boundary. Therefore, we use a calibrated version of a OCSVM model, using goals from $\mathcal{R}$ as the prior distribution:

$$p_{\text{valid}}(g) \propto \mathbb{P}(g \in \tilde{\mathcal{G}}|\mathcal{R}) = R_\psi(g), \tag{5}$$

where $R_\psi(g)$ is the output of the OCSVM model, with parameters $\psi$, and $\tilde{\mathcal{G}}$ is the set of states that the agent is or was able to reach until the current iteration.

As the agent progresses and expands its set of achieved states through training, it eventually reaches the environment boundary. In this case, $p_{\text{valid}}(g) \propto \mathbb{P}(g \in \tilde{\mathcal{G}}|\mathcal{R}) \approx \mathbb{P}(g \in \mathcal{G}|\mathcal{R})$.

The pseudo-code of SVGG is given in Algorithm 1. After a random initialization of the particles set $\Omega$, the approach collects data by running the agent with sampled goals corresponding to particles from $\Omega$, which are used to feed learning of success predictor $D_\phi$ and reachability predictor $R_\psi$. From

---

**Algorithm 1** *RL with Stein Variational Goal Generation*

---

1: **Input:** a GCP $\pi_\theta$, a parameterizable environment $\mathcal{M}$, a number of particles $m$, a success predictor $D_\phi$, a reachability predictor $R_\psi$, buffers of transitions $\mathcal{B}$, reached states $\mathcal{R}$ and success outcomes $O$, a kernel $k$, numbers $n$, $t^{(r)}$, $t^{(m)}$, $t^{(p)}$, $l^{(r)}$, $l^{(m)}$ and $l^{(p)}$.
2: Sample a set of particles $\Omega = \{x^i\}_{i=1}^m$ uniformly from states reached by pre-runs of $\pi_\theta$;
3: **for** $n$ epochs **do**                                                                  ▷ *Data Collection*
4:   **for** $t^{(r)}$ iterations **do**
5:     Sample a batch $\{g^i\}_{i=1}^{l^{(r)}}$ from $\Omega$;
6:     **for** $i$ from 1 to $l^{(r)}$ **do**
7:       $\tau \leftarrow$ Rollout $\pi_\theta(.|., g = g^i)$;
8:       Store all $(s_t, a_t, s_{t+1}, r_t, g^i)$ from $\tau$ in $\mathcal{B}$ and every $s_t$ from $\tau$ in $\mathcal{R}$;
9:       Optionally (HER): Store relabeled transitions from $\tau$;
10:       Store outcome $(g^i, I(s_{|\tau|} \approx g^i))$ in $O$;
11:     **end for**
12:   **end for**
13:   **for** $t^{(m)}$ iterations **do**                                                      ▷ *Model Update*
14:     Sample a batch $\{(g^i, s^i)\}_{i=1}^{l^{(m)}}$ from $O$;
15:     Update model $D_\phi$ (e.g., via ADAM), with gradient of $\mathcal{L}_\phi$ (3));
16:   **end for**
17:   Update model $R_\psi$ according to all states in $\mathcal{R}$;                          ▷ *Prior Update*
18:   **for** $t^{(p)}$ iterations **do**                                                      ▷ *Particles Update*
19:     Compute the density of the target $p_{\text{goals}}$ for the set of particles $\Omega$ using 2;
20:     Compute transforms: $\phi^*(x_i) = \frac{1}{m} \sum_{j=1}^m \left[ k(x_j, x_i) \nabla_{x_j} \log p_{\text{goals}}(x_j) + \nabla_{x_j} k(x_j, x_i) \right]$;
21:     Update particles $x_i \leftarrow x_i + \epsilon \phi^*(x_i), \quad \forall i = 1 \cdots n$;
22:   **end for**
23:   Improve agent with any Off-Policy RL algorithm                                          ▷ *Agent Improvement*
      (e.g., DDPG) using transitions in $\mathcal{B}$;
24: **end for**

---

these adjusted models, $p_{goals}$ is used to compute transforms of the particles in $\mathcal{S}$. At the end of each epoch, an off-Policy RL algorithm[2] is used to improve the agent.

**Theorem 1.** ***Recovery property**: Let us denote as $\mathcal{G}^+$ the set of goals $g$ such that $R_\psi(g) > 0$ and $\mathcal{C} \in \mathbb{R}^d$ its convex hull. Assume that, at a given iteration $l$, $D_\phi(g) \approx 1$ for every $g \in \mathcal{G}^+$ (i.e., the whole set $\mathcal{G}^+$ is considered as mastered by $D_\phi(g)$), and that, on that set, $R_\psi$ is well calibrated: $R_\psi(g) \approx P(g \in \mathcal{G})$. Assume also that we use a kernel which ensures that the Kernelized Stein Discrepancy $KSD(q, p_{goals})$ of any distribution $q$ with $p_{goals}$ is 0 only if $q$ weakly converged to $p_{goals}$[3]. Then, with no updates of the models after iteration $l$ and a number of particles $m > 1$, any area $\omega \subseteq \mathcal{G}^+ \cap \mathcal{G}$ with diameter $diam(\omega) \geq \sqrt{d} \frac{diam(\mathcal{C})}{(\sqrt[d]{m}-1)}$ eventually contains at least one particle, whenever $KSD(\{x^i\}_{i=1}^n, p_{goals}) = 0$ after convergence of the particle updates.*

The above theorem (proof in Appendix A) ensures that, even if the success model over-estimates the capacities of the agent for some area $\omega$ (e.g., due to changes in the environment, catastrophic forgetting or success model error), some particles are likely to go back to this area once every goal in $\mathcal{G}$ looks well covered by the agent, with an increasing probability for more particles. This way, the process can reconsider over-estimated areas, by sampling again goals in them, and hence correcting the corresponding predictions, which leads to attracting attention of $p_{skills}$ back to these difficult areas. Approaches such as MEGA do not exhibit such recovery properties, since they always sample at the boundary of $\tilde{\mathcal{G}}$, which is likely to incrementally grow towards $\mathcal{G}$. The particles of our approach can be seen as attention trackers which remodel the behavior distribution and mobilize the effort on useful areas when needed. This is much better than uniformly sampling from the whole space of achievable states with small probability, which would also ensure some recovery of forgotten areas but in a very inefficient way. This theoretical guarantee of SVGG is empirically validated by the

---

[2]We use DDPG (Lillicrap et al., 2015) in our experiments

[3]As assumed for instance in Liu (2017). This is not always true, but gives strong insights about the behavior of SVGD. Please refer to Gorham & Mackey (2017) for more discussions about KSD.

experiment from Figure 4, which shows the good recovering property of our approach, after a sudden change in the environment.

# 4 EXPERIMENTS

To measure SVGG's performance and compare it to baselines, we evaluate the resulting policy on the entire space of valid goals $\mathcal{G}$ in our environment. We consider a success coverage metric, defined as $S(\pi) = \frac{1}{|\mathcal{G}|} \int_{\mathcal{G}} \mathbb{P}(\pi \text{ reaches } g) dg$. The goal space being continuous, we evaluate the policy on a finite uniform subset $\hat{\mathcal{G}} \subset \mathcal{G}$. Then our objective reduces to:

$$S(\pi) = \frac{1}{|\hat{\mathcal{G}}|} \sum_{i=1}^{|\hat{\mathcal{G}}|} \mathbb{P}(\pi \text{ reaches } g_i) = \frac{1}{|\hat{\mathcal{G}}|} \sum_{i=1}^{|\hat{\mathcal{G}}|} \mathbb{E}_{\tau \sim \pi}[\mathbb{1}\{\exists s \in \tau, \min_{s^* \in \mathcal{S}_{g_i}} ||s - s^*||_2 < \delta\}]. \quad (6)$$

To build $\hat{\mathcal{G}}$, we split $\mathcal{G}$ into areas following a regular grid, and then uniformly sample 30 goals inside each part of the division.

We investigate the following questions: 1) Does SVGG maximize the success coverage objective compared to recent intrinsic motivation work? 2) What target difficulty (i.e., beta distribution as described above) is more efficient to maximize coverage? 3) Is SVGG robust to environment changes? 4) Is SVGG Robust to catastrophic forgetting?

Our main results are based on a Point-Maze environment: an agent moves a point without mass within a 2d maze with continuous spaces of states and actions. Due to the walls, the goal space presents many discontinuities. In the absence of expert knowledge, introduced via reward shaping or a given curriculum, the success coverage objective is very hard to maximize. To conduct our experiments, we rely on the modular RL code base (Pitis et al., 2020a).

## 4.1 COMPARED APPROACHES

As baselines, we choose two methods from the literature that span the existing trends in unsupervised RL with goal conditioned policies (GCP), MEGA Pitis et al. (2020b) and Goal-GAN Florensa et al. (2018). In addition, the Random baseline randomly selects the behavior goal among past achieved states. We also perform two SVGG ablations: a *No Prior* version, which considers $p_{goals} \propto p_{skills}$ and a *Only Prior* version, where $p_{goals} \propto p_{valid}$. All compared approaches use DDPG and HER with a mixed strategy described in the appendix. In every version of SVGG, we use $n = 100$ particles to approximate the target distribution. Implementation details about all considered architectures are given in Appendix C.

## 4.2 MAIN RESULTS

Our main results in Figure 3 shows the success coverage of baselines over 4 different maze environments. SVGG is the only method that discovers and reaches all valid goals in 4M agent steps in all considered environments.

Due to the known stability issues in GAN training, Goal-GAN is the least efficient of baselines. Another explanation of the failure of Goal-GAN is that the generator is likely to output goals outside the valid space, which is not the case for MEGA or SVGG. MEGA chooses the behavior goal in a Replay Buffer of achieved goals, while the prior distribution $p_{valid}$ considered in SVGG keeps particles inside valid areas. Nevertheless, we observe that Goal-GAN performs better in environments with linear structures, such as Mazes 3 and 4. On the other hand, it has a hard time to generate goals of intermediate difficulty when the structure is random, as in Mazes 1 and 2.

The minimum density heuristic of MEGA efficiently discovers all feasible goals in the environment, but our results show that its success plateaus in almost all the considered environments. MEGA's intrinsic motivation only relies on state novelty. Thus, when the agent has discovered all feasible states, it is unable to target areas that the agent has reached in the past but has not mastered yet. Figure 3 also reports results on two classical environments: AntMaze and a hard version of FetchPickAndPlace. SVGG also outperforms MEGA on AntMaze, although the considered maze is simpler than those considered in our main experiments, but still induces a goal space with discontinuities involved by

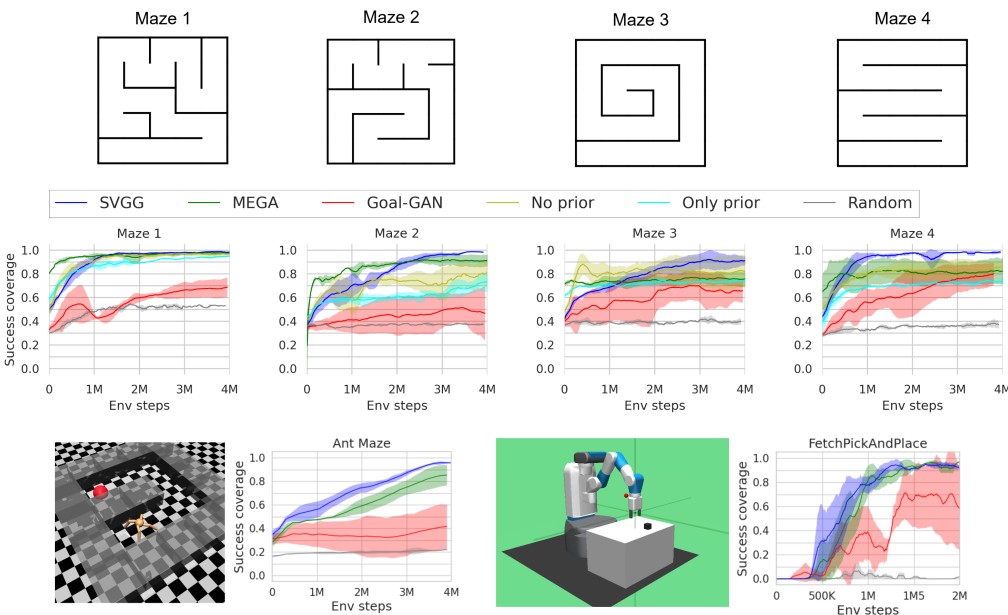

Figure 3: Success coverage evaluated over 4 different PointMazes, AntMaze and FetchPickAndPlace (4 seeds each). SVGG outperforms recent intrinsic motivation algorithms (MEGA and Goal-GAN), as well as naive baselines.

walls. On FetchPickAndPlace, where the goal space is fully smooth, our approach obtains comparable results to its competitors.

**Recovery Property** In Figure 4, we demonstrate the advantages of SVGG's intrinsic motivation over MEGA's in a changing environment. Walls are suddenly added in the mazes during the training process, after the methods had solved the entire goal space (dot red line from the figure). Curves show that MEGA is unable to fully adapt to the new maze setting, which corresponds to Maze 1 from Figure 1 (although MEGA solved Maze 1 in 1M steps in that experiment with constant conditions). On the other hand, SVGG is able to discover new difficulties due to the environment modification and focuses on those to finally solve the entire maze in less than 1.5M steps.

We also observe that the advantages of our method over MEGA in terms of recovery ability are more significant when changes in the environment are more drastic (i.e., when starting from maze A).

**SVGG analysis** To gain further intuition on how SVGG maximizes the success coverage, we show in Figure 5 the evolution of the particles throughout training. As the agent progresses and reaches novel and harder goals, the $D_\phi$ model updates its predictions. Thus, the target distribution $p_{\text{goals}}$ is updated accordingly (background color of the $2^{nd}$ row of the figure). The particles $q = \{g_i\}_{i=1}^n$ are then moved toward new areas of intermediate difficulty through SVGD to minimize $KL(q||p_{\text{goals}})$.

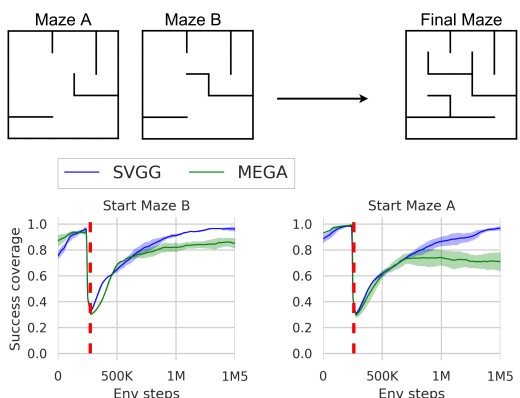

Figure 4: Success coverage evolution in a changing environment for MEGA and SVGG (4 seeds each).

Figure 5 also highlights the recovery property of SVGG. When the model detects that the agent has nothing else to learn in the environment, $p_{\text{goals}}$ reduces to $p_{\text{valid}}$, that is at this point uniform on the entire goal space. Therefore, the particles spread uniformly over the environment and thus prevent

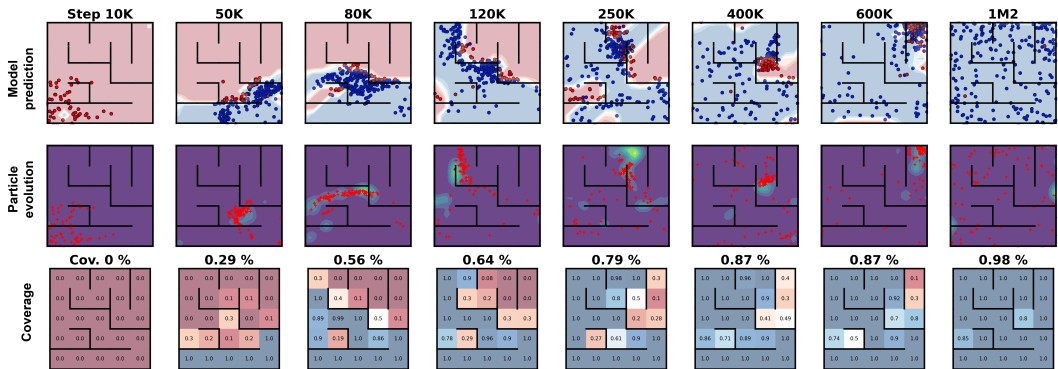

Figure 5: Parallel visualization of the model's prediction ($1^{st}$ row), the particles evolution ($2^{nd}$ row) and of the coverage ($3^{rd}$ row) throughout training in Maze 1.

SVGG from catastrophic forgetting, as the model rediscovers areas that the agent has forgotten how to reach (cf. rightmost column in Figure 5). Additional analyses on SVGG are given in C.2.

## 5 CONCLUSION

This paper introduces a new approach for multi-goal reinforcement learning in deterministic environments. Our algorithm, SVGG, leverages Stein Variational Gradient Descent to monitor a model w.r.t. its goal reaching capabilities. Using this model, the agent addresses goals of intermediate difficulty, resulting in an efficient curriculum for finally covering the whole goal space. Moreover, SVGG can recover from catastrophic forgetting by means of SVGD particle optimization, which is a classic pitfall in intrinsically motivated RL.

Studying the impact of the number of particles is left for future work. Actually, the target distribution being in constant evolution, the KL divergence minimization objective is hard to reach at all times, which makes it difficult to claim that using more particles is always better. Furthermore, a previous work D'Angelo & Fortuin (2021) spotted exploration failures in SVGD, and suggests that periodically annealing the attraction force in particle optimization (1) is required to enable particles to cover non-trivial distributions, e.g. in multimodal settings or in high dimensions.

Some limitations should be addressed in future work. The environments used in our experiments have low dimensional goal space, which facilitates learning the particles and the agent's model. When the agent's observation and goal space will be images, the agent should learn a compact latent space of observation like in Pong et al. (2019), Yarats et al. (2021) and Liu & Abbeel (2021), using various representation learning techniques like contrastive learning, prototypical representations or variational auto-encoders. In future work, such procedures could be added to SVGG in order to learn a latent goal space from observations, where both particle and model's skills learning could be performed. This would result in an end-to-end algorithm learning to discover and reach all possible goals in its environment from pixels.

Besides, we also envision to address larger and stochastic environments, where additional uncertainty estimation should be added to the goal generation process, to prevent the agent getting stuck in uncontrollable states (like a TV screen showing white noise) like in Chua et al. (2018) and Pathak et al. (2019). Methods such as model disagreement between multiple agents could be explored in future work.

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

## A    PROOF OF THEOREM 1

**Theorem 1.** *Recovery property: Let us denote as $\mathcal{G}^+$ the set of goals $g$ such that $R_\psi(g) > 0$ and $\mathcal{C} \in \mathbb{R}^d$ its convex hull. Assume that, at a given iteration $l$, $D_\phi(g) \approx 1$ for every $g \in \mathcal{G}^+$ (i.e., the whole set $\mathcal{G}^+$ is considered as mastered by $D_\phi(g)$), and that, on that set, $R_\psi$ is well calibrated: $R_\psi(g) \approx P(g \in \mathcal{G})$. Assume also that we use a kernel which ensures that the Kernelized Stein Discrepancy $KSD(q, p_{goals})$ of any distribution $q$ with $p_{goals}$ is 0 only if $q$ weakly converged to $p_{goals}$[4]. Then, with no updates of the models after iteration $l$ and a number of particles $m > 1$, any area $\omega \subseteq \mathcal{G}^+ \cap \mathcal{G}$ with diameter $diam(\omega) \geq \sqrt{d}\frac{diam(\mathcal{C})}{(\sqrt[d]{m}-1)}$ eventually contains at least one particle, whenever $KSD(\{x^i\}_{i=1}^n, p_{goals}) = 0$ after convergence of the particle updates.*

*Proof.* In settings where the KSD of a distribution $\mu$ from a target $p$ is 0 only if $\mu$ weakly converged to $p$, Liu (2017) previously proved that, for any compact set, the empirical measure $\mu^n$ of $\mu$, computed on a set of $n$ particles, converges weakly towards the target $p$ when a sufficient number of particles is used. Thus, under our hypotheses, the set of particles of SVGG appropriately represents $p_{goals}$ after a sufficient number of steps of Stein transforms.

Now, we know that particles cannot leave $\mathcal{G}^+$ since $R_\psi = 0$ outside, and so does $p_{goals}$. Since $R_\psi$ is well calibrated on $\mathcal{G}^+$, we also know that $R_\psi = 1$ on every area of $\mathcal{G}^+$ only containing achievable goals. Thus, since $p_{skills} = 1$ in $\mathcal{G}^+$, $p_{goals}$ is maximal and constant for any area $\omega \in \mathcal{G}^+ \cap \mathcal{G}$. This implies that the concentration of particles in any area of $\mathcal{G}^+ \cap \mathcal{G}$ is greater than if particles were uniformly spread over $\mathcal{C}$. In that case, for any particle $x$ from the set $\{x_i\}_{i=1}^m$, we know that $P(x \in \omega | KSD(q = \{x_i\}_{i=1}^m, p_{goals}) = 0) \geq P(x \in \omega | KSD(q = \{x_i\}_{i=1}^m, U(\mathcal{X})) = 0)$, with $\mathcal{X}$ an hypercube of $d$ dimensions with side length equal to $diam(\mathcal{C})$ and $U(\mathcal{X})$ the uniform distribution over $\mathcal{X}$.

Next, if $KSD(q = \{x_i\}_{i=1}^m, U(\mathcal{X})) = 0$, we know that particles are spread as a grid over each dimension of $\mathcal{X}$. Thus, in each dimension of $\mathcal{X}$ any particle is separated from its closest neighbor by at most a difference of $diam(\mathcal{C})/(\sqrt[d]{m} - 1)$ in the worst case. Thus, any area $\omega$ with diameter greater than $\sqrt{d}\frac{diam(\mathcal{C})}{(\sqrt[d]{m}-1)}$ is guaranteed to contain a particle in that case, which concludes the proof.

$\square$

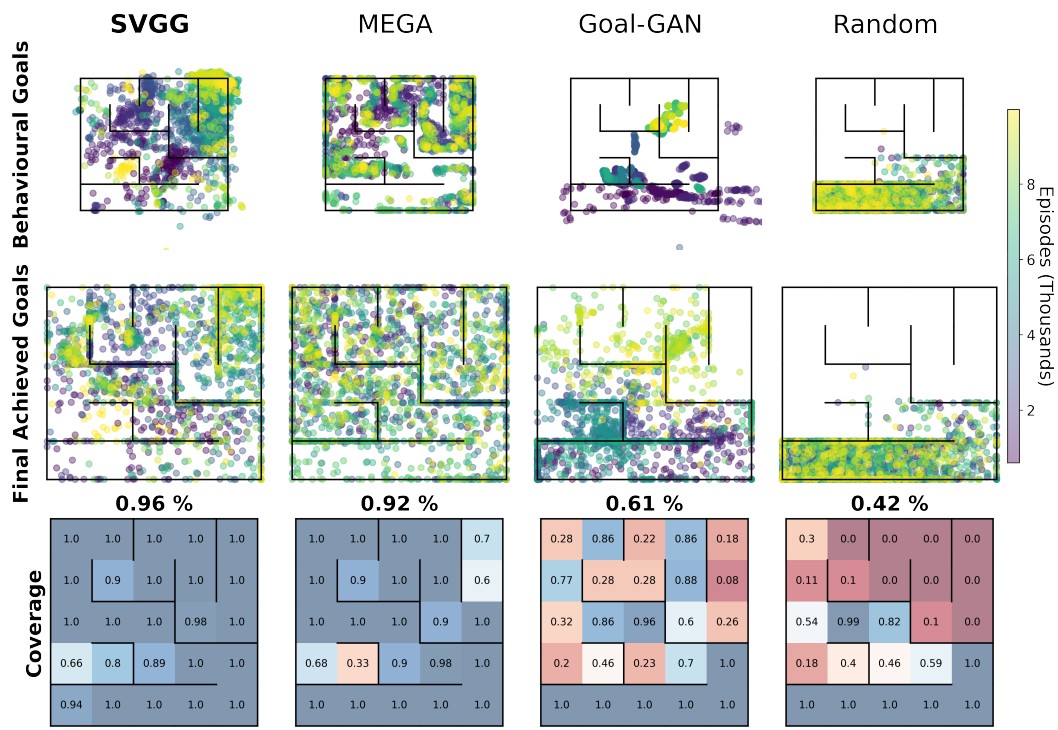

Figure 6: Visualization of behavioral goal ($1^{st}$ row), achieved goals ($2^{nd}$ row) and success coverage ($3^{rd}$ row) for 1M steps in the Maze 1 environment.

## B    VISUALIZATION OF GOALS

We visualize behavioral and achieved goals in Maze 1 (Figure 6) and Maze 2 (Figure 7), in parallel with the success coverage after training. The main advantages of our method lie in the capacity to target difficult areas and avoid catastrophic forgetting, which results in nearly optimal success coverage. We observe that MEGA efficiently discovers the environment but is unable to detect where the agent has not mastered yet a goal in order to target it. This also leads to catastrophic forgetting and a lack of adaptation when the environment changes, a case studied in the main paper.

One can see that the generation of GOIDs in Goal-GAN is very unstable and tricky in such discontinuous goal space, especially as the generator is susceptible to output goals outside the environment boundary, which SVGG avoids with the prior distribution.

## C    ADDITIONAL EXPERIMENTS AND IMPLEMENTATION DETAILS

### C.1    ENVIRONMENTS

**Pointmaze**    We use a 2D pointmaze environment where the state space and goal space are (x, y) coordinates (the agent cannot see the walls), and the agent moves according to its action, which is a 2-dimensional vector with dimensions constrained to $[-0.95, 0.95]$. All methods are tested on 4 $5 \times 5$ mazes with different shapes and a highly discontinuous goal space. The maximum number of steps in one trajectory is set to 30. We argue that the difficulty of these environments does not lie in their size, but rather in the complexity of their goal space and thus of the trajectories adopted by the agent to reach these goals.

---

[4]As assumed for instance in Liu (2017). This is not always true, but gives strong insights about the behavior of SVGD. Please refer to Gorham & Mackey (2017) for more discussions about KSD.

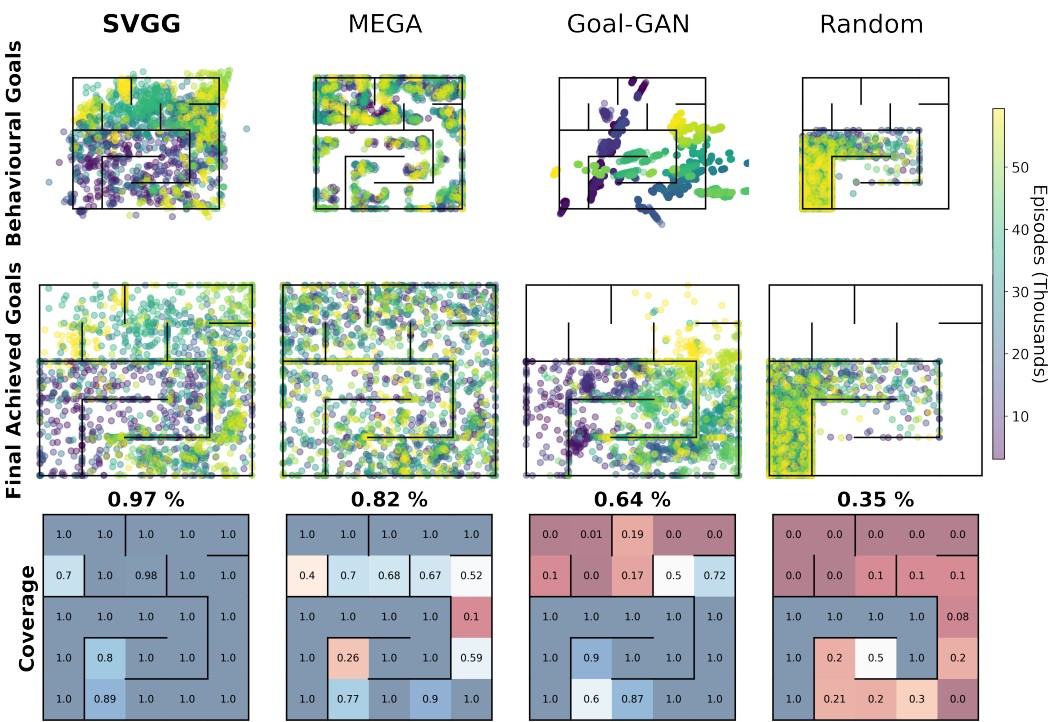

Figure 7: Visualization of behavioral goal ($1^{st}$ row), achieved goals ($2^{nd}$ row) and success coverage ($3^{rd}$ row) for 4M steps in the environment on Maze 2. Our method (SVGG) is the only one to achieve nearly perfect success coverage.

**Antmaze** An Ant has to move around in a U-shape hallway maze of size 20x4, the goal space remains (x,y) coordinates of the Ant as in Pointmaze. However, the exploratory behavior is much simpler than in the considered pointmazes environments, as the maze is simply U-shaped. The difficulty lies in the control of the Ant dynamics, as it must move its legs with a 8-dimensional action space and the observation space is a 30-dimensional vector corresponding to the angles of the legs.

**Fetch Pick and Place (Hard version)** To test the versatility of our method, we conduct additional experiments on a hard version of FetchPickAndPlace-v1 from OpenAI gym Brockman et al. (2016). The agent controls a robotic arm which must pick and place a block to a 3D desired location. In the hard version, the target goals are placed between 20 and 45cm in the air, while in the original version, 50% of target goals are on the table and the remaining ones are placed between 0 and 45cm in the air.

The distance to reach the goals in all environment is $\delta = 0.1$. While this environment presents a greater difficulty in terms of control (the action dimension is 4 and the state dimension is 10 which correspond to the various positions of the robot joint), the goal generation component is significantly easier as the goal space is smooth, there is no obstacle for the agent to bypass. As Figure 3 shows, SVGG solves the environment within 1.5M steps, but does not significantly outperform MEGA.

We argue that the interest of our goal generation algorithm resides in environments with highly discontinuous goal space as the action control is largely supported by the RL algorithm (e.g. DDPG). Therefore, the smaller difference between MEGA and SVGG in this environment was expected, as SVGG is mainly designed to target non-smooth goal spaces, and to avoid pitfalls such as catastrophic forgetting. Experiments on environments that are complex both in terms of goals and control are left for future work.

## C.2 CONTROL OF THE SAMPLING DIFFICULTY

Using beta distributions reveals which goal difficulty an intrinsic motivation algorithm should aim at for an agent to efficiently explore and control an environment. In Figure 9, we compare 5 versions

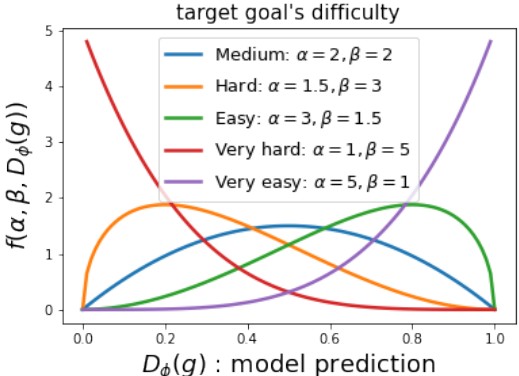

Figure 8: Modulation of goal's target difficulty with beta distributions. $D_\phi(g)$ being the model's predicted probability of the agent reaching the goal $g$.

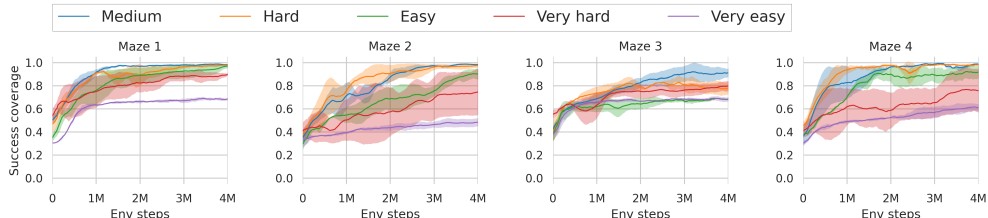

Figure 9: Success coverage evolution for 5 target difficulties as intrinsic motivation for SVGG, on 4 mazes and 4 seeds each.

of SVGG using different distributions from Figure 8, on the 4 previously considered mazes. One can observe that extreme targets difficulties are the least effective ones, especially the *Very easy*, which is too conservative to efficiently explore new areas of the space. On the other hand, SVGG performs very well with *Medium* and *Hard* distributions. This suggests that the optimal goal difficulty is somewhere between medium and hard.

## C.3 BASELINES

The authors of MEGA Pitis et al. (2020b) train a GCP with previously achieved goals from a replay buffer. Their choice of goals relies on a minimum density heuristic, where they model the distribution of achieved goals with a KDE. They argue that aiming at novel goals suffices to efficiently discover and control the environment. We use the original implementation of the authors, the pseudocode is described in Algorithm 2.

---

**Algorithm 2** *MEGA*

---

1: **Input:** a GCP $\pi_\theta$, a parameterizable environment $\mathcal{M}$, reached states $\mathcal{R}$, a KDE model $P_{as}$ of the achieved states, numbers $c$, $n$, $N$, $t^{(r)}$ and $l^{(r)}$.
2: Initialize $\mathcal{R}$ with states reached by pre-runs of $\pi_\theta$;
3: **for** $n$ epochs **do**                                                   ▷ *Data Collection*
4:     **for** $t^{(r)}$ iterations **do**
5:         Sample a batch $\{g^i\}_{i=1}^N$ uniformly from $\mathcal{R}$;
6:         Eliminate unachievable candidates if their $Q$-values are below cutoff $c$;
7:         Choose goals $\{g^i\}_{i=1}^{l^{(r)}} =_{g_i} P_{as}(g_i)$
8:         **for** $i$ from 1 to $l^{(r)}$ **do**
9:             $\tau \leftarrow$ Rollout $\pi_\theta(.|., g = g^i)$;
10:            Store all $(s_t, a_t, s_{t+1}, r_t, g^i)$ from $\tau$ in $\mathcal{B}$ and every $s_t$ from $\tau$ in $\mathcal{R}$;
11:            Optionally (HER): Store relabeled transitions from $\tau$;
12:            Store outcome $(g^i, I(s_{|\tau|} \approx g^i))$ in $O$;
13:        **end for**
14:    **end for**
15:    Update KDE model $P_{as}$ with uniform sample from $\mathcal{R}$;         ▷ *KDE Update*
16:    Improve agent with any Off-Policy RL algorithm                           ▷ *Agent Improvement*
           (e.g., DDPG) using transitions in $\mathcal{B}$;
17: **end for**

---

**Algorithm 3** *Goal-GAN*

---

1: **Input:** a GCP $\pi_\theta$, a parameterizable environment $\mathcal{M}$, a goal Generator $G_{\theta_g}$, a Discriminator $D_{\theta_d}$, a success predictor $D_\phi$, buffers of transitions $\mathcal{B}$, reached states $\mathcal{R}$ and success outcomes $O$, numbers $n$, $P_{min}$, $P_{max}$, $t^{(r)}$, $l^{(r)}$, $l^{(m)}$, $k^{(m)}$, $l^{(g)}$ and $k^{(g)}$.
2: Initialize $G_{\theta_g}$ and $D_{\theta_d}$ with pre-runs of $\pi_\theta$;
3: **for** $n$ epochs **do**                                                   ▷ *Data Collection*
4:     **for** $t^{(r)}$ iterations **do**
5:         Sample noise $\{z_i\}_{i=1}^{l^{(r)}} \sim \mathcal{N}(0, 1)$
6:         generate $\{g^i\}_{i=1}^{l^{(r)}} = G_{\theta_g}(\{z_i\}_{i=1}^{l^{(r)}})$
7:         **for** $i$ from 1 to $l^{(r)}$ **do**
8:             $\tau \leftarrow$ Rollout $\pi_\theta(.|., g = g^i)$;
9:             Store all $(s_t, a_t, s_{t+1}, r_t, g^i)$ from $\tau$ in $\mathcal{B}$ and every $s_t$ from $\tau$ in $\mathcal{R}$;
10:            Optionally (HER): Store relabeled transitions from $\tau$;
11:            Store outcome $(g^i, I(s_{|\tau|} \approx g^i))$ in $O$;
12:        **end for**
13:    **end for**
14:    Sample a batch $\{(g^i, s^i)\}_{i=1}^{l^{(g)}}$ from $O$;                 ▷ *GAN training*
15:    Label goals (GOID or not) with model $D_\phi$ : $\{y_{g_i}\}_{i=1}^{l^{(g)}} = \{P_{min} < D_\phi(g_i) < P_{max}\}_{i=1}^{l^{(g)}}$
16:    **for** $k^{(g)}$ iterations **do**
17:        Update $G_{\theta_g}$ and $D_{\theta_d}$ (e.g. with ADAM) with gradients of LSGAN losses; (7)
18:    **end for**
19:    **for** $k^{(m)}$ iterations **do**                                      ▷ *Model Update*
20:        Sample a batch $\{(g^i, s^i)\}_{i=1}^{l^{(m)}}$ from $O$;
21:        Update model $D_\phi$ (e.g., via ADAM), with gradient of $\mathcal{L}_\phi$ (agent model loss described in the paper);
22:    **end for**
23:    Improve agent with any Off-Policy RL algorithm                           ▷ *Agent Improvement*
           (e.g., DDPG) using transitions in $\mathcal{B}$;
24: **end for**

---

Goal-GAN Florensa et al. (2018) uses a procedural goal generation method based on GAN training. As our SVGG, it aims at sampling goals of intermediate difficulty, which they define as $\mathcal{G}_{\text{GOID}} = \{g | P_{min} < P_\pi(g) < P_{max}\}$, $P_\pi(g)$ being the probability for the policy $\pi$ to reach goal $g$, $P_{min}$ and $P_{max}$ are hyperparameters. A Discriminator $D_{\theta_d}$ is trained to distinguish between goals in $\mathcal{G}_{\text{GOID}}$ and

other goals, while a generator $G_{\theta_g}$ is trained to output goals in $\mathcal{G}_{\text{GOID}}$ by relying on the discriminator outputs. They optimize $G_{\theta_g}$ and $D_{\theta_d}$ in a manner similar to the Least-Squares GAN (LSGAN) with the following losses

$$
\begin{aligned}
\min_{\theta_d} V(D_{\theta_d}) &= \mathbb{E}_{g \sim \mathcal{R}} \left[ y_g (D_{\theta_d}(g) - 1)^2 + (1 - y_g)(D_{\theta_d}(g) + 1)^2 \right] \\
\min_{\theta_g} V(G_{\theta_g}) &= \mathbb{E}_{z \sim \mathcal{N}(0,1)} \left[ D_{\theta_d}(G_{\theta_g}(z))^2 \right],
\end{aligned}
\tag{7}
$$

where $y_g$ is the label that indicates whether $g$ belongs to $\mathcal{G}_{\text{GOID}}$ or not. In Florensa et al. (2018), the authors use Monte-Carlo sampling of the policy to estimate $y_g$. For efficiency reasons, we use a learned model of the agent's capabilities as in SVGG. The pseudocode is described in Algorithm 3.

## C.4 HINDSIGHT EXPERIENCE REPLAY

The original goal relabeling strategy introduced in HER Andrychowicz et al. (2017) is the *future*, which consists in relabeling a given transition with a goal achieved on the trajectory later on. This is very effective in sparse reward setting to learn a GCP. However, many works suggested that relabeling transitions with goals outside the current trajectory helps the agent generalize across trajectories. For example, one can use inverse RL to determine the optimal goal to relabel a given transition Eysenbach et al. (2020). We use a naive version of this method. As in Pitis et al. (2020b), we relabel transitions for DDPG optimization using a mixed strategy. All methods presented in this work use the same strategy.

10% of real experience are kept while 40% of the transitions are relabeled using the *future* strategy of HER. We relabel the remaining 50% transitions with goals outside of their trajectory, with randomly sampled goals among the past behavioral and achieved goals. The latter part of the strategy helps share information between different trajectories that often contains similar transitions.

## C.5 ARCHITECTURES AND HYPERPARAMETERS

| Hyper-Parameter | Value |
|---|---|
| **DDPG** | |
| Batch size for replay buffer | 2000 |
| Discount factor $\gamma$ | 0.99 |
| Action L2 regularization | 1e-1 |
| Action noise maximum std (Gaussian noise) | 0.1 |
| Warm up steps before training | 2500 |
| Actor learning rate | 1e-3 |
| Critic learning rate | 1e-3 |
| Target Network update fraction for soft parameters update update | 0.05 |
| Activation for actor and critic networks | Gelu |
| Actor dense layers sizes | (512, 512, 512) |
| Critic dense layers sizes | (512, 512, 512) |
| Replay buffer size | max training steps |
| | |
| **SVGD** | |
| Number of particles $m$ | 100 |
| Optimization every k agent's steps | 20 |
| Number of particle moves for one optimization $k^{(p)}$ | 1 |
| Bandwidth $\sigma$ for RBF kernel $k(.,.)$ for RKHS | 1 |
| Learning rate $\epsilon$ | 1e-3 |
| | |
| **Agent's skill model $D_\phi$** | |
| Dense layers sizes | (64, 64) |
| K gradient steps per optimization | 10 |
| Learning rate | 1e-3 |
| Training batch size $l^{(m)}$ | 100 |
| Training history length (trajectories) | 1000 |
| Optimize every (steps) | 4000 |
| Number of training steps $k^{(m)}$ | 100 |
| Activations | Gelu |
| | |
| **OCSVM prior** | |
| Bandwidth $\sigma$ for RBF kernel $k(.,.)$ for RKHS | 1 |
| Optimize every (agent's steps) | 4000 |
| | |
| **MEGA** | |
| Bandwidth of KDE RBF kernel | 0.1 |
| KDE optimize every (agent's steps) | 1 |
| Number of state samples for KDE optimization | 10000 |
| Number of sampled candidate goals $N$ | 100 |
| $Q$-value cutoff $c$ | -3 |
| | |
| **Goal-GAN** | |
| Generator input dimension (gaussian noise) | 4 |
| Generator dense layers sizes | (64, 64) |
| Discriminator dense layers sizes | (64, 64) |
| Optimize every (agent's steps) | 2000 |
| GAN training batch size $l^{(g)}$ | 200 |
| Number of GAN steps in for one optimization $k^{(g)}$ | 100 |
| Discriminator/Generator Learning rate | 1e-3 |
| $P_{min}$ minimum probability for GOID | 0.1 |
| $P_{max}$ maximum probability for GOID | 0.9 |

### C.6 RESSOURCES

Every seed of each experiment was run on 1 GPU in the following list {Nvidia RTX A6000, Nvidia RTX A5000, Nvidia TITAN RTX, Nvidia Titan Xp, Nvidia TITAN V}. We estimate the amount of training time to be 4.000 hours, most of it executed in parallel, as we had full time access to 12 GPUs.

