# OpenReview forum: "Stein Variational Goal Generation for adaptive Exploration in Multi-Goal Reinforcement Learning"
_ICLR.cc/2023/Conference — Submitted to ICLR 2023_

### Official Review · Reviewer_ZXEG · 2022-10-24

**Confidence:** 3
**Clarity, Quality, Novelty And Reproducibility:** The paper is clearly presented, but t…
**Correctness:** 3
**Technical Novelty And Significance:** 2
**Empirical Novelty And Significance:** 2
**Recommendation:** 5

**Strength And Weaknesses:**

The paper is well-written and the idea is simple but novel. However, I have some concerns and thus do not recommend acceptance.

Q1. SVGG seems to be the direct combination of stein variational gradient and goal generation, which weakens the novelty of the paper.

Q2. Recently, many related works [1-7] have been proposed to solve goal-reaching tasks. Some of them [4-6] learn a goal space capturing the inter-state reachability, which can address the difficulty of discontinuities in the goal spaces. More discussions about the comparison between SVGG and these methods are required.

Q3. The environments used in the experiment section seem to be simple. How well does SVGG perform on RL tasks of complex environments?

References:

[1] Huang, Z., et al. Mapping state space using landmarks for universal goal reaching. NeurIPS, 2019.

[2] Ghosh, Dibya, et al. "Learning to reach goals via iterated supervised learning." ICLR 2021.

[3] Hoang, Christopher, et al. "Successor Feature Landmarks for Long-Horizon Goal-Conditioned Reinforcement Learning." Advances in Neural Information Processing Systems 34 (2021): 26963-26975.

[4] Zhang, Tianren, et al. "Generating adjacency-constrained subgoals in hierarchical reinforcement learning." NeurIPS, 2020.

[5] Zhang, Tianren, et al. “Adjacency constraint for efficient hierarchical reinforcement learning”, IEEE TPAMI, 2022.

[6] Chane-Sane, et al. "Goal-conditioned reinforcement learning with imagined subgoals." ICML, 2021.

[7] Kim, Junsu., et al. Landmark-guided subgoal generation in hierarchical reinforcement learning. NeurIPS, 2021.


**Summary Of The Paper:**

The paper proposes a novel approach Stein Variational Goal Generation (SVGG) to build on recent automatic curriculum learning techniques to address the difficulty of discontinuities in the state or goal spaces for goal-reaching tasks. Experiments show that SVGG achieves state-of-the-art results for hard-exploration RL tasks in several environments with low-dimensional state spaces and deterministic transition models.

**Summary Of The Review:**

I have some concerns and thus do not recommend acceptance.

---

> ### Author Response · Authors · 2022-11-18
> **Response to reviewer ZXEG**
>
> We thank the reviewer for their comment. We are glad that the reviewer finds the paper well-written and the idea simple and novel. Below we address their concerns and react to the proposed references.
>
> - **Novelty**
>
> The reviewer considers SVGG as the direct combination of SVGD and goal generation, which weakens the novelty of our paper. We respectfully disagree.
> Our main contribution is to model the goal distribution with SVGD, resulting in a generation process that combines both an intermediate difficulty criterion with $p_{\text{skills}}$ and a novelty criterion with the exploration of particles that is fostered by the repulsive force of SVGD. With this combination, we have the best of both worlds between pure exploration methods such as MEGA and methods that only pursue intermediate difficulty without any interest for goals novelty like GoalGAN. The recovery property that prevents from catastrophic forgetting results from using this combination, and is also a contribution of our work.
>
> Besides, we thank the reviewer for the references to other papers from the goal-conditioned RL literature. However, they do not fit with our context, which is multi-goal RL, they use prior knowledge on the environment, and they assume to have access to the desired goals. Those goals being long-horizon, they use various techniques such as graph learning or generation of subgoals to decompose the finals tasks into several easier ones.
> In our setting, we do not have access to such relevant tasks in the environment. This corresponds to many real world scenarios where we first need to explore the whole environment. SVGG explores and learns to reach every goal in the environment. However, in the same unsupervised setting, we could envision to first explore the environment with a method such as a goal exploration process (see e.g. [1]), and then learn to efficiently reach them with techniques that are present in the spotted references (see e.g. [2]).
>
> - **Experiments**
>
> About the difficulty of our test environments, mazes like the ones used in our experiments are harder than they seem at a first glance, because the agent is blind to walls and cannot anticipate the many discontinuities of the goal space. Second, in terms of success coverage, mazes are very difficult tasks, as they require the policy to learn diverse behaviors that correspond to the many discontinuities in the goal space.  Our focus in this paper is on addressing the difficulties raised by such discontinuities. By contrast, more complex environments in terms of observation and action space such as FetchPickandPlace put all the stress on the RL algorithm, which is not our focus. Nevertheless, we see that SVGG still outperforms the baselines in AntMaze.
>
> [1] Colas, C., Sigaud, O., and Oudeyer, P. Y. (2018, July). GEP-PG: Decoupling exploration and exploitation in deep reinforcement learning algorithms. In International conference on machine learning (pp. 1039-1048). PMLR.
>
> [2] Campos, V., Trott, A., Xiong, C., Socher, R., Giró-i-Nieto, X., and Torres, J. (2020, November). Explore, discover and learn: Unsupervised discovery of state-covering skills. In International Conference on Machine Learning (pp. 1317-1327). PMLR.

---

### Official Review · Reviewer_ipWS · 2022-10-24

**Confidence:** 3
**Correctness:** 1
**Technical Novelty And Significance:** 3
**Empirical Novelty And Significance:** Not applicable
**Recommendation:** 3

**Clarity, Quality, Novelty And Reproducibility:**


### Clarity comments

In theorem 1: I understand $D\rightarrow 1$ as $D\approx 1$, but if the authors meant taking a limit (wrt. what?), I'd appreciate it if they clarified it in the paper. Furthermore, $v$ is not defined in the main paper (only in the proof).

It's not entirely	clear for me how the updated probabilities $p\_{}goals$ are actually used. Are they estimated over each particle from $\Omega$ at the beginning of each epoch and then used during sampling in "Agent improvement"? Is sampling for training $D$ and $R$ also following the distribution?

### Evaluation
The paper is mostly understandable, although hard to follow in places when a reader is not that familiar with SVGD. The method seems novel, but the experimental results (and the weaknesses above) are not encouraging.

The claims in the paper are strong: "SOTA results for hard-exploration RL problems", "(their method) actively adapts goals to useful areas, even with a poor model of success", "SVGG can be considered as a tradeoff between classical GCRL and evolutionary methods", which I don't think are substantiated.

I find the work reproducible.

**Strength And Weaknesses:**

## Strengths

On the high level, the mechanism of storing tasks in the buffer and attaching weights to them based on some criteria to maximize learnability makes sense.

Development of models which encourage similarity to already solved tasks as a method of preventing overestimation is an interesting idea (even though I'd love it to happen in a single model, see below).

## Weaknesses

### Splitting D and R
D and R have a similar role: to estimate the density for whether a task is good for training. I don't find the motivation presented for splitting them (that R distinguishes "unachievable" tasks) convincing, as the tasks the paper tests on: PointMaze, and AntMaze don't have such tasks.

It seems to me that the actual role of R is to correct for overestimation behavior of D, but it should be possible to combine them into a single model (eg. with loss being a combination of conservative R and value-accurate D), which would make the method simpler.

Furthermore, the theorem giving intuition about why the method works assumes that R behaves exactly like D (is 0 where $D<0$ and 1 elsewhere).[1]


### Catastrophic forgetting
The main advantage of the work over MEGA the authors seem to stress is the ability of SVGG to escape catastrophic forgetting, as no goal will ever be "forgotten" when it's already learned, but will be continuously revisited. Overall, that's a desiderata that is simple to satisfy by just uniformly sampling previously achieved goals with some fixed probability $\alpha$. In fact, the MEGA paper[2] proposed such a variant, calling it OMEGA (where they also estimate $\alpha$). It seems appropriate to compare against that version of their work.

Furthermore, I don't understand why $D=1$ would not lead to catastrophic forgetting for SVGG. As $q$, expressed by the particles means to approximate $p\_{}goals = exp(f(D)) * R$, and $f(1) = 0$ for most of the tested $f$s (which makes most sense, we don't want to be oversampling solved tasks), the weight of tasks with $D=1$ would be $0$, even if they are present in $\Omega$, leading to catastrophic forgetting.

### Zone of Proximal Development
Choosing $\alpha$ and $\beta$ fixed over training corresponds to prioritizing tasks of a particular level of difficulty (percentage of being solved). Eg. authors' choice of $\alpha=\beta$ corresponds to tasks with $D=1/2$ having the highest weight throughout the training, even though at the beginning the distribution of task difficulty (as measured by $D$) may be centered around 0.01 (meaning that tasks with $D=1/2$ would be way too hard, or most likely: having their difficulty overestimated).

This seems unintuitive, as the Curriculum Learning methods typically attempt to sample tasks of medium difficulty for the agent's current abilities, and requires a further explanation.

### Exploration

Despite the claim about working in non-Euclidean spaces, R seems to be strongly discouraging exploration: if the initial set of particles would be chosen in a deep corner of the task-space (because the initial policy is not able to explore well), the method will have to slowly go through the goals based on what OCSVM predicts as the boundary of the dataset, even if the policy, after a couple of steps, would be able to explore much wider and generate the new particles covering the space more uniformly.

### Experiments
The experiments are not convincing enough. The domains are simpler than, e.g. the ones shown in MEGA (smaller mazes, less tasks). There are two baselines: one of them is 4 years old, and another one a the simpler version of the method presented in the MEGA paper. Ideally, I would like to see a comparison with 2>= current methods, eg. SPaCE [3] and OMEGA or fixed, $\alpha>0$ version of MEGA.

Despite the maze tasks (on which the improvement is shown) are simpler than the ones used in MEGA, the reference scores of MEGA are much worse (see their figure 3 in [3]), suggesting that something is wrong with the experimental protocol.

## Suggestions

I would like to see a discussion on why not embed the tasks in a better metric space than the Euclidean one, as it seems easier than the proposed method.

[1] By the way, it seems that the proof says that $R=1$ on whole $G^+$, but the theorem assumes $R=1$ only on $\tilde{G}$.

[2] [Maximum Entropy Gain Exploration for Long Horizon Multi-goal Reinforcement Learning](https://arxiv.org/pdf/2007.02832.pdf)

[3] [Self-Paced Context Evaluation for Contextual Reinforcement Learning](https://arxiv.org/pdf/2106.05110.pdf)

**Summary Of The Paper:**

Authors describe an automatic curriculum learning method for goal-based (reward is to reach a goal or not) multi-task RL they call SVGG. It keeps a set of current goals as particles, estimates their density using Stein Variational Gradient Descent, and keeps updating the particles to keep presenting interesting challenges to the agent without leaving already-learned goals behind to prevent catastrophic forgetting.

The estimation of the density relies on two models: D, behaving like a value function, estimating success rate for reaching a particular goal, and R, a more conservative model, not allowing to sample goals far away from what was achieved so far (with the motivation behind R being to not allow sampling invalid tasks).

Authors motivate the work with HER being prone to local optima when using Euclidean distance as the metric between goals which live in a discontinuous space.

Authors claim their approach could be easily adapted to settings where goals don't correspond to states to be reached. I would like the authors to expand on their statement, as it's not obvious for me how to extend the work.

Authors compare their method to MEGA and GoalGAN on 4 mazes from PointMaze, AntMaze,, and robotic-arm task FetchPickAndPlace.

**Summary Of The Review:**

The paper presents a new method. I have numerous concerns about its ability to meet the claims made in the paper, and the experimental results are not encouraging. Before these issues are addressed, I discourage publishing the paper at ICLR.

---

> ### Author Response · Authors · 2022-11-18
> **Response to reviewer ipWS part 2/2**
>
> - **Comparison with OMEGA**
>
> Then the reviewer suggests comparing against OMEGA rather than MEGA. OMEGA is a version of the MEGA algorithm which incorporates prior knowledge on the environment goal space, which is out of the scope of our work. Therefore, MEGA is not a simpler version of OMEGA but rather they are two different algorithms with different objectives. In the OMEGA setting, the authors assume to have access to a desired goal distribution $p_{dg}$, from which they start to sample goals at probability $\alpha$ once the achieved goals distribution $p_{dg}$ and $p_{ag}$ start to overlap. In their work, $\alpha = 1/\max (b + D_{\text{KL}(p_{dg} || p_{ag}}),1)$, with $b=-3$, they are sure to achieve $\alpha = 1$ at convergence. As a consequence of only sampling goals in $p_{dg}$ at convergence, whose support is a small subset of the goal space, OMEGA is prone to catastrophic forgetting. By contrast, in our work we assume no prior knowledge on the environment. Thus MEGA is a more appropriate baseline than OMEGA.
>
> We also would like to stress that in our setting, uniformly sampling from the achieved goal distribution is not trivial. One could use Maximum Entropy Prioritization [1] directly on $p_{ag}$ to sample from $\mathcal{U}(\text{support(}p_{ag}))$, which is computationally expensive, and it would require an additional parameter $\alpha$ as in OMEGA to adjust the probability of sampling from this distribution. We found our method simpler in comparison, as SVGD automatically shifts toward sampling from $\mathcal{U}(\text{support(}p_{ag}))$ with the model $R$ when $D$ does not detect any more goals of intermediate difficulty.
>
>
> - **Various clarifications**
>
> When $D = 1$, we get $p_{\text{goals}(g)} \propto exp(f(D))*R \approx exp(0)*R = R$, and so the target distribution of the particles to fit is the uniform distribution over the valid goal space, which enables SVGG to recover some forgotten goals.
>
> About the zone of proximal development, the reviewer's comment reveals that some aspects of our approach have been misunderstood.
> Since the set of particles is optimized for approximating a Boltzmann distribution, their spread is driven by the relative (not absolute) values of D and R over the support. At the beginning of the process, $D \approx 0$ almost everywhere. Thus particles stay close to the starting point of the agent. Then, some of the tasks directly connected to the starting point are solved with small probability. Those tasks are favored until others get a better score of $f$. Even with alpha=beta, it is true that favored tasks are those with 0.5 success rate, tasks with e.g. only 0.01 are prioritized ones if no other task gets better rate.
>
> Please refer to the appendix Fig. 9, for experimental results on the impact of the shape of $f$, which shows that setting $\alpha=\beta$ is the best choice.
>
>
> - **Exploration**
>
> We agree that $R$ limits the exploration behavior, but is also avoiding uncontrolled exploration outside the environment boundaries. Hitting the right trade-off is needed here. The objective with $R$ is to slowly extend the particles boundaries as the agent makes progress, which fosters a controlled exploration. However, the trajectories of the agent are not limited in space by $R$, only the particles are, thus it does not prevent the agent from generating new trajectories and then exploring its environment progressively during training. $R$ is in fact less conservative than $D$ in many cases, as learning to master goals is more difficult than discovering new areas during agent rollouts.
>
>
> - **Experiments**
>
> The reviewer also points that the scores of MEGA are much worse than in the original paper, which is true as we do not use the same evaluation metrics. The authors in MEGA use a very small subset of the goal space which corresponds to the desired goal distribution $p_{dg}$ for evaluation. With the coverage success metric, we evaluate MEGA on the all valid goal-space, which makes more sense for our context where we do not use prior knowledge on what are the interesting tasks in the environment.
>
> [1] Zhao, R., Sun, X., and Tresp, V. (2019, May). Maximum entropy-regularized multi-goal reinforcement learning. In International Conference on Machine Learning (pp. 7553-7562). PMLR.

---

> > ### Comment · Reviewer_ipWS · 2022-11-21
> > **Thank you for explanations**
> >
> > I took note of all of your responses. I only answer the parts where there seems to be some further disagreement / misinterpretation:
> >
> > - **On splitting D and R**
> >
> > Authors argue that splitting D and R is necessary to accommodate the general experimental setting where there are unsolvable tasks in the environment. It was my understanding that this is not the case for the experiments performed, ie. while the method itself doesn't assume that there are either solvable or unsolvable tasks present, the actual tasks themselves are always solvable (ie. within the maze frontier). Could the authors clarify that?
> >
> > If all the tasks presented to the agent are indeed always solvable, yet the performance of the model drops when R is not present (which, as I understand, is equivalent to the perfect R, always predicting that the task is solvable), the presented motivation behind splitting D and R seems inadequate. In the review I proposed a hypothesis for explaining that behavior (R is to correct for overestimation of D) which would match the data and would suggest that a single model could be possible.
> >
> > - **On catastrophic forgetting**
> >
> > The authors present an argumentation on why their method may be better than uniformly sampling already solved goals. I find the intuitive argumentation not satisfactory; and that's why I was calling for an epsilon-greedy baseline. Authors' work bear a significant complexity compared to, eg. MEGA, and as such I consider it reasonable to analyze simpler settings which satisfy the same desiderata (resampling already solved tasks).
> >
> > I don't understand why authors claim that uniformly sampling from the achieved distribution is difficult. Why can't a buffer of old tasks be stored and uniformly sampled from?
> >
> > - **On Zone of Proximal Development**
> >
> > I find that your comment doesn't address my concern. Let me rephrase. Let's ignore R (assume it always returns 1), assume $\alpha = \beta$ and imagine that at the beginning of training, we have a number of tasks with varying values of D from 0.005 to 0.02. The proposed method will be prioritizing tasks with the currently highest solve rates (the easiest ones), 0.02. This will be happening until the middle of the training, when the behavior will change to choosing the hardest ones.
> >
> > In my mind, the ZPD says that we should be choosing tasks of medium complexity taking into account the current abilities of the agent, ie. the tasks with performance ~0.01 in the scenario above. The assumption that the tasks whose current expected solve rate is 0.5 (or any other, fixed one) are best without consideration for the solve rates of other tasks seems to be limiting.

---

> > > ### Author Response · Authors · 2022-11-22
> > > **Answer to the remaining questions**
> > >
> > > Thank you for your response, here are our answers to the remaining issues:
> > >
> > > - **Splitting D and R**
> > >
> > > The reviewer is right about the fact that in our maze experiments, all goals inside the maze frontiers can be solved. However, without any knowledge of the frontiers at training time and without $R$, the particles would not be constrained to stay inside the boundaries, some of them would spread outside the frontiers where goals are unfeasible.
> > >
> > > With that in mind, the objective of $R$ isn't only to correct the overestimation of $D$, but also to provide a distribution predicting the validity of the goals. We stand by the observation that merging this objective into $D$ would result in a complicated trade-off as $D$ would need to predict the difficulty of the goals together with their validity, though these are very different pieces of information.
> > >
> > > - **Catastrophic forgetting**
> > >
> > > Two different things are not to be confused: uniform sampling of previous tasks within the replay buffer as the reviewer mentioned and uniform sampling on the support of valid goals. The first one corresponds to our _Random_ baseline, that only resamples previous achieved goals during training. This uniform sampling on the replay buffer is biased by the policy behavior, for example if the achieved goal distribution $D_{ag}$ is Gaussian, rarely the method will resample goals at the tails of $D_{ag}$, as you can see in Figure 7 on the Appendix for the  _Random_ baseline.
> > >
> > > On the other hand, uniform sampling on the support of $D_{ag}$, i.e sampling from  $U(\text{support}(D_{ag}))$, is much more interesting as we try to eliminate the bias toward the policy behavior. However, this is not trivial, for example [1] skew the distribution with importance sampling, and [2] sample goals w.r.t their inverse likelihood (learned with Gaussian Mixture) to prioritize low density tasks.
> > >
> > > We could incorporate such methods so as to arbitrarily resample previous tasks from $U(\text{support}(D_{ag}))$. However, this would add complexity to our approach and is not necessary as uniform sampling of $U(\text{support}(D_{ag}))$ is performed automatically when $p_{\text{goals}} = R$, i.e. when $D=1$ on the entire space.
> > >
> > > That being said, our main argument is that uniform sampling over the valid goal space like in MEGA is not enough to optimize the success coverage, which can be seen in our experiments. Our method is able to focus on intermediate difficulty areas most of the time and switch to uniform sampling when needed ($D=1$) to avoid catastrophic forgetting. This process is much more efficient than sampling from $U(\text{support}(D_{ag}))$ arbitrary even if the time is not right, i.e when the agent has already found an interesting area to focus on that will provide a strong learning signal.
> > >
> > >
> > > - **Zone of Proximal Development**
> > >
> > > Our hypothesis is that the zone of proximal development is around $D(g)=0.5$, which we call _medium difficulty_. There seems to be a confusion about this notion, the _medium difficulty_ does not correspond to the median value of all predicted probabilities, but rather correspond to values that are the closest to $0.5$.
> > >
> > > Taking your example, $D$ computes the agent's current abilities and outputs that it has a probability up to $0.02$ to reach some tasks, then the tasks that correspond to $0.02$ are closer to $0.5$ and thus have a better chance of providing useful learning signal.
> > >
> > >
> > > [1] Pong, V., Dalal, M., Lin, S., Nair, A., Bahl, S., Levine, S. (2020). Skew-fit: State-covering self-supervised reinforcement learning. In Proc. of ICML, Vol. 119, pp. 7783–7792.
> > >
> > > [2] Zhao, R., Sun, X., and Tresp, V. (2019, May). Maximum entropy-regularized multi-goal reinforcement learning. In International Conference on Machine Learning (pp. 7553-7562). PMLR.

---

> ### Author Response · Authors · 2022-11-18
> **Response to reviewer ipWS part 1/2**
>
> We thank the reviewer for the constructive review with many suggestions and for spotting a minor error in the proof of our theorem. Below we answer some of the reviewer's concerns and clarify some points where we believe the reviewer may have misunderstood us.
>
> The reviewer would like to know how we would extend our work to the case where the goal space differs from the state space.
> In fact we already do this for AntMaze and Fetch environments (contrary to what we claimed in the background section, thanks for pointing this mistake): in that cases $S_g$ corresponds to the set of states whose spatial coordinates are the same as those of the specified goal. In our work, one only needs a goal-satisfaction function that takes a state or a set of states as input and determines whether the goal is satisfied with this input. Thus, the only requirement is the availability of an Oracle that maps a continuous goal representation to the set of success states. This is now specified in the paper.
>
> - **Splitting D and R**
>
> The reviewer also suggests that $D$ and $R$ have similar roles and should not be separated, and do not fully understand why $R$ is needed, as PointMaze and AntMaze do not present unfeasible goals.
>
> We would like to emphasize that at training, SVGG does not incorporate prior knowledge on the environment. For example in PointMaze and AntMaze, all goals outside the maze frontier are unfeasible, therefore we have to keep the particles inside the frontiers considering we don't know them at first. Since we assume that we do not know the frontier, we have to construct incrementally a prior distribution that will output the probability that a goal is valid or not, that is the role of $R$.
>
> On the other hand, $D$ enables to target goals of intermediate difficulty, which is a different objective. Merging it with the validity check objective using a combination of losses would result in a very hard compromise to achieve. We aim at sampling goal with highest uncertainty of success, but with high probability of validity. We believe learning two separate components with simple objectives and architecture (a standard MLP for $D$ and a One Class SVM for $R$) is the best choice in our context. We remind the reviewer that the impact of $D$ and $R$ have been studied in the ablations that we conduct in the experiments in Figure 3, the result shows that both component are needed to get higher results.
>
> Regarding the theorem, the reviewer points that R and D behave exactly the same. This is not true in the general case. In the theorem, we consider the extreme case where every achievable goal is considered as mastered by the success model, to illustrate that when this happens, the particles go back in already considered areas, by spreading uniformly on the set. This ensures to reconsider predictions for that areas, and focus on some of them if necessary (in case of catastrophic forgetting or change in the environment). But the reviewer is right, there was some minor mistakes in the theorem, we fully rewrote it along with its proof  (the conclusion remains unchanged).
>
> - **Catastrophic Forgetting**
>
> The reviewer questions about the relevance of SVGG regarding the recovery property, compared to epsilon-greedy to would sample from a mixture with a uniform distribution. This is true that it would allow to reconsider forgotten areas of goals. But only very punctually: it would only provide useful goals by chance, but would not help re-focus attention on that area that probably requires more work than a unique training sample. Our particles can be seen as attention trackers, able to remodel the behavior distribution and mobilize the effort on useful areas.
> Rather than uniformly sampling goals to give training data to the agent (besides possibly in areas that became too difficult for the agent, from which no training signal could be leveraged), our particles scour the space in search of interesting areas on which to focus, with the guarantee in the long run of being able to re-distribute themselves over the entire space, ensuring that they will refocus on areas that may have been forgotten (or after a change in the environment).
>
> Please refer to the end of Section 3 for a new discussion on that particular point.

---

### Official Review · Reviewer_BQng · 2022-10-25

**Confidence:** 5
**Clarity, Quality, Novelty And Reproducibility:** The paper is really well written and …
**Correctness:** 3
**Technical Novelty And Significance:** 2
**Empirical Novelty And Significance:** 3
**Recommendation:** 3

**Strength And Weaknesses:**

Overall I think the paper presents an interesting new technique of sampling goals. The idea is presented clearly and the evaluations are easy to follow. However I do have some concerns.


### Strength

The proposed method is clearly presented and well motivated. I find the idea of weighting the desired goal probability with the intermediate difficulty and prior distribution very intuitive, and I can imagine a class of different methods building on top of this idea. I also find the ablation study and visualization of the proposed method is very informative. The behavior of the proposed method can be directly understood from the generated goals.


### Weaknesses

From my understanding of the paper, it seems like the core novelty of the paper is the particular goal distribution jointly weighted by the intermediate difficulty and goal prior. This objective is intuitive as it gives high weight to valid goals that have intermediate difficulty that the policy can only only some of the times. However, it is one of many other objectives that encourages similar things (for example, the GOID objective from Goal-GAN), so it is not clear to me why this particular choice is better. It would be important for the authors to justify the objective either theoretically or empirically.


Moreover, once the desired goal distribution is formulated, it seems to me that SVGD is just a particular choice of generative models used to learn this goal distribution. Given an unnormalized probability density, many other techniques, such as MCMC, Langevin dynamics and variational inference, can be used to generate samples from this distribution, and some of them are much simpler than SVGD. Especially in the low-dimensional goal space that the authors experiment with, numerical integration can be accurately calculated to normalize the goal density. The mentioned recovery property can also be obtained by simply mixing the learned goal distribution with a uniform prior distribution.  Hence, it is not clear to me why SVGD is a good choice here.


The empirical performance of the proposed method is also not very strong. Among the domains the authors experiment with, only in 3 out of 6 domains the proposed method outperforms prior methods significantly.


**Summary Of The Paper:**

This paper focuses on the problem of goal conditioned reinforcement learning (GCRL), where the authors propose a new way of sampling goals during the data collection phase. Specifically, the authors propose to sample goals with probability proportional to two factors: the prior distribution of goals and the intermediate difficulty of goals. The intermediate difficulty probability is measured by a beta distribution which is parameterized by a learned classifier to give high weight on goals that the agent can achieve only some of the time. This ensures that the generated goals stay within the valid regions, and are neither too easy nor too difficult. The authors then propose to learn this distribution via Stein variational gradient descent (SVGD) to obtain a goal generator model.

The authors evaluate the proposed method empirically on maze, ant maze and simulated robot pick and place environments, and the results suggest that the proposed method outperforms prior methods in terms of goal success rate and sample efficiency.


**Summary Of The Review:**

While I think the paper presents an interesting new technique of sampling goals, I believe that the proposed method is not sufficiently justified and the empirical performance is not very strong. Hence I cannot recommend acceptance of this paper in its current state.

---

> ### Author Response · Authors · 2022-11-18
> **Response to reviewer BQng part 2/2**
>
>  - **Catastrophic Forgetting**
>
> The reviewer questions about the relevance of SVGG regarding the recovery property, compared to epsilon-greedy to would sample from a mixture with a uniform distribution. This is true that it would help reconsider forgotten areas of goals. But only very punctually: it would only provide useful goals by chance, but would not help re-focus attention on areas requiring more work than a unique training sample.
> Our particles can be seen as attention trackers, able to remodel the behavior distribution and mobilize the effort on useful areas.
> Rather than uniformly sampling goals to give training data to the agent (besides possibly in areas that became too difficult for the agent, from which no training signal could be leveraged), our particles scour the space in search of interesting areas on which to focus, with the guarantee in the long run of being able to re-distribute themselves over the entire space, ensuring that they will refocus on areas that may have been forgotten (or after a change in the environment).
>
> Please refer to the end of Section 3 for a new discussion on that particular point.
>
> - **Experiments**
>
> Also, the reviewer is not convinced by the superiority of SVGG over baselines, considering that SVGG only outperforms these baselines in 3 out of 6 environments. Our point in the paper is not that SVGG will outperform the baselines in any multi-goal environment, but rather to better understand in which contexts it does so. With the results, we see that SVGG does not outperform MEGA when the environment is too simple (Maze1) or when the difficulty lies in the size of the observation and action space (FetchPickandPlace) but it does so all the more when the environment is more difficult in terms of discontinuities in the goal space.
>
> - **Justifications**
>
> The reviewer concludes by saying the proposed method is not sufficiently justified, but this contradicts their previous statements that "The proposed method is clearly presented and well motivated. I find the idea of weighting the desired goal probability with the intermediate difficulty and prior distribution very intuitive”. We hope that clarifications we made in the paper and this answer will help to clarify the justification of the approach.

---

> > ### Comment · Reviewer_BQng · 2022-11-30
> > **Response to Authors**
> >
> > First of all I'd like to thank the authors for the detailed response. Here are my comments.
> >
> > ### Difference with GoalGAN And Bayesian Approximation
> > First of all I want to clarify that I'm not concerned about the similarity to the GOID objective in the GoalGAN paper. After all, I think that most effective goal proposal methods will make use of similar objectives in order to sample goals that maximizes the learning signal. What I'm really concerned about this paper is the fact that using which distribution to sample goals from and using what generative model to learn and sample from this distribution are essentially two problems that should be studied separately. In this paper, the authors chose a particular goal distribution and use a particular generative model to perform goal proposal, but it is not clear to me whether the choice of goal distribution or the choice of generative model is providing the advantage. Therefore, in order to support the claim that both of these choices are crucial, it would be necessary to compare to some other goal distributions using the same generative model, and some other generative models using the same goal distribution. I think the latter is probably more important, since SVGD is not necessarily the most easy to use generative model and other simpler methods could be used in these low dimensional goal spaces.
> >
> >
> > ### Experiments
> > My concern here is that the experiments really don't demonstrate the advantage of the proposed method. To demonstrate that SVGG really works better than prior methods, more experiments are needed.
> >
> > Overall, I'd like to keep my evaluation of the paper.

---

> > > ### Author Response · Authors · 2022-12-07
> > > **Answer to the remaining issues**
> > >
> > > We agree that both the choice of generative modeling and goal generation objective should be argued.
> > > For the objective part, we especially choose a flexible objective (with the beta distributions) that allow us to conduct experiments on various target difficulty of goals, which is very informative on the validity of our hypothesis of proximal development. Also, the ablations (without prior, only prior) inform us on the key component of our objective.
> > >
> > > On the generative part, we stand by the fact that SVGD is the most appropriate among all MCMC / Variational Inference methods, because:
> > >
> > > - we don't need to compute the intractable normalization function (that is also intractable in 2D because we don't use the mazes frontiers).
> > >
> > > - We don't need to specify a variational distribution family, which provide more modeling expressiveness that is crucial in the context of high discontinuities such as in mazes.
> > >
> > > - The optimal transform is computed deterministically, which is key to the interpretation of the particles movement and allow us to derive the recovery property in theorem 1. This would not have been possible with any others modeling tools or if we just used direct sampling from the replay buffer such as in MEGA.
> > >
> > > That being said we agree that more modeling tools should be studied, this is the reason we choose as baselines Goal-GAN and MEGA that uses respectively GANs and direct sampling from the replay buffer.

---

> ### Author Response · Authors · 2022-11-18
> **Response to reviewer BQng part 1/2**
>
> We thank the reviewer for their comments. We are glad to read that the reviewer found the paper "really well written" and that "the proposed method is easy to follow". Below we answer the reviewer's main concerns.
>
> - **Difference with GoalGAN**
>
> The reviewer is right to highlight a similarity with GoalGAN as the objective present in $p_{\text{skills}}$ is very similar to the one of GoalGAN. We aim at sampling goals of intermediate difficulty similarly to the GOIDs of GoalGAN. However, their GOIDs objective: $\{g | \alpha <\mathbb{P}^{\pi}(g) < \beta \}$ is not differentiable. One can see $p_{\text{skills}}$ as a smooth version of the GoalGAN objective whose differentiability is leveraged in SVGD (where we need to compute $\nabla_{g}logp(g)$ for optimization). This differentiability helps obtain the recovery property due to the repulsive force that fosters exploration.
>
> Besides, SVGD offers us a novel way to sample new goals (not only buffer goals as in MEGA). Moving in the support space of goals rather than generating goals from scratch via a network as in GoalGan results in a much more stable approach, successfully modeling the whole distribution landscape that we approximate. In fact, GoalGAN usually tends to move far from the set of achievable states, and has no mechanism to foster reconsideration of already mastered areas of the space. We clarified this in the introduction of the paper.
>
>
> - **Bayesian Approximation**
>
> Besides, the reviewer states that, given an non-normalized probability density, many other techniques, such as MCMC, Langevin dynamics and variational inference, could be used to generate samples from the goal distribution, and some of them are much simpler than SVGD. The reviewer is right to mention that SVGD is one method among others to performs Bayesian inference. One emblematic method is MCMC with for example the Metropolis-Hastings algorithm. Although MCMC methods converge asymptotically to the exact distribution, they are known to be very slow. This is not suitable in our case where the goal distribution is a moving target.
> Classical variational inference methods need a normalized target distribution to compute the Evidence Lower Bound, and then directly minimize the KL divergence between the approximate and the true distribution.
> This also implies the difficult choice of the family of target distributions, which would be very limitative for such irregular space.
> On the other hand, Stochastic Langevin Dynamics and Stein Variational Gradient Descent do not directly minimize the KL divergence between the approximation and the target. Both iteratively find the optimal vector field to move the particles toward this objective, which allows them to work on the derivative of the KL divergence and then with unnormalized distributions. Therefore, both methods can be considered in our setting.
>
> To our knowledge, the current state of the research on variational inference does not clearly favor one method over the other, although several experimental studies [1], [2] show that SVGD marginally outperforms SGLD (Stochastic Gradient Langevin Dynamics). SVGD benefits from computing the exact optimal transport direction in a deterministic way, while Langevin dynamics requires a random diffusion term such as a Brownian process.
> For our setting where we are interested in tracking useful areas to train the agent, dealing with a set of particles that represent the state of the landscape appears better fitted than only relying on a single one that produces samples very correlated in time. This choice also offers a better interpretability of the process, by providing comprehensive images of the current behavior distribution at any time of the training.
>
> Please finally note that, even in a low dimensional space, as our method does not use prior knowledge on the environment, we cannot directly normalize the goal distribution on the support of the valid goal space as the latter is unknown. Normalization could be done on the convex hull of observed states, but this would be very inefficient for non convex state spaces.
>
> [1] Qiang Liu and Dilin Wang. Stein variational gradient descent: A general purpose Bayesian inference algorithm. arXiv preprint arXiv:1608.04471, 2016.
>
> [2] Wei-Cheng Chang, Chun-Liang Li, Youssef Mroueh, and Yiming Yang. Kernel Stein generative modeling. arXiv preprint arXiv:2007.03074, 2020.

---

### Official Review · Reviewer_TSmE · 2022-10-25

**Confidence:** 3
**Correctness:** 2
**Technical Novelty And Significance:** 2
**Empirical Novelty And Significance:** 2
**Recommendation:** 5

**Clarity, Quality, Novelty And Reproducibility:**

- The paper is not written clearly and it is difficult to fully understand the contributions of the work. There are at times too much details, but all appears convoluted in some way, making the primary contributions not explicit.

- In terms of novelty, it may be that there are some components of the work that are novel (although my understanding from reading the paper is that it is mostly adapting from prior works). Most importantly, I think the proposed approach is overly complicated to achieve a rather achievable solution to the stated problems. Therefore, I am concerned about the practical use case of the work and its impact, if it gets accepted to a venue like ICLR.

-It maybe that the paper is written in a way that the contributions were not clear enough; otherwise, the difficulty of the algorithm in the way it is derived - my primary concern is the reprodocubility of the work (not in terms of the results, but if someone actually tries to re-implement the proposed approach)

**Strength And Weaknesses:**

	- This paper addresses the important problem of exploration in goal spaces, since to design a curriculum of goal generation for better generalization, it is important to learn a model that can help with exploration of new goals. The work is primarily based on context of automatic goal curriculum generation
	- One minor thing about the paper in general though is it provides too much not connected information in the write-up, making it hard to fully understand the context of the paper. It would be better if the paper could be written more clearly, with the main contributions focussed on. From the introduction itself, it was not clear what are the primary contributions of this work.
	- The work adapts from using SVGD in context of approximating the goal distribution for curriculum generation. Section 2.2 providing context of this, and stating out related work that uses SVGD is useful to see.
	- Section 3 in my opinion could have been written more clearly; and figure 1 could be explained better. The overall approach seems too complicated for actually being impactful for a general community. The work primarily focuses on hard exploration problem, and my understanding is that SVGD based approach is required for approximating a goal distribution, such that this learned model can then be used for novel goal generation?
	- I do not understand the context of equation 2? Why is the distribution of goals related to skills and valid goals? It would be helpful if authors could provide more context of why the distribution of goals that we are trying to approximate, indeed relies on these two? The text below equation (2) provides some context to this, and p_valid (g) is understandable; but how does the learned skills also depend on p(g) is perhaps not clear? My understanding is, we want to approximate goals but at the same time make sure these goals are achievable based on the learned skills? Is this the correct way to interpret this?
	- Equation 4 and 5 then provides a tractable approach to approximating these two terms, which are again dependent on learning two separate models?
	- Could the authors explain the coverage metric in more details? It seems like there is an approximation being made for this coverage metric? The problem seems intractable, and is the same as trying to approximate the state distribution for coverage?
Experimental results in my opinion does not seem entirely convincing. The most difficult task seems to be the FetchPickandPlace and it is not clear if the improvements are significant, especially given the fact that the overall proposed model in this paper seems overly complicated (see below)

**Summary Of The Paper:**


This work considers a multi-goal learning setting where the agent learns across a diverse set of goals by building its own representation model of goal space. The work tackles exploration in a multi-goal learning setting since being able to learn across diverse goals during train plays a vital role for generalization. However, since the goal space requires exploration - the work proposes a new mechanism based on stein variational goal generation to tackle the goal exploration problem.


**Summary Of The Review:**


Please see above - I think the contributions of the work were not clear enough and the overall model is too complicated to be of actual impact.

---

> ### Author Response · Authors · 2022-11-18
> **Response to reviewer TSmE**
>
> We thank the reviewer for their comments about the paper.
>
> - **Clarity and Positioning**
>
> The main point of the reviewer is that our work is lacking clarity. Even if the other reviewers rather positively assess our work in that respect, the comments of the reviewer helped us realize that our positioning and some more technical elements needed to be clarified. To address this point, we completely rewrote the abstract and the introduction as well as the first lines of Section III and a few other paragraphs, and we updated Fig. 1.
>
> - **Positioning w.r.t. Related Works**
>
> In terms of positioning, it should now be clearer that SVGG improves both over methods like MEGA which optimizes achieved goal coverage but fails to reliably reach the corresponding behavior goals and methods like GoalGAN which generates a distribution of goals of intermediate difficulty, but do so in an unstable way and fails to explore efficiently.
>
> Besides, we hope that Eq. (2) is now clearer: $p_{\text{goals}}$ is the evolving distribution of goals, or curriculum, that we are looking for, $p_{\text{skills}}$ results from sampling the goals with the highest entropy according to a learned model of the goal reaching capabilities of the agent, and $p_{\text{valid}}$ is a learned distribution which acts as a prior and intends to prevent the agent from addressing unreachable goals.
>
>
>
> The reviewer also wonders whether the approach is not too complicated. We believe this impression was due to insufficient writing clarity. SVGG only has three simple components, a one class SVM to learn the $p_{valid}$ distribution of valid goals, a neural network trained to predict goal reaching success and a set of particles trained with SVGD to target the goals where the prediction of success is closest to 0.5. Then the agent just samples a goal by selecting one of the particles. We hope that our rewriting effort will change the opinion of the reviewer about this apparent complexity.
>
>
> In terms of reproducibility of the work, all hyper-parameters are provided in the Appendix and the code will be made publicly available as specified in footnote 1.
>
> - **Coverage Metric**
>
> The reviewer is also right to ask about the success coverage metrics, which is one of our contributions but was not enough put forward in the previous version of the paper. The success coverage measures the average success of the agent on all possible goals in the environment. To compute this success coverage metrics, we need to draw uniform samples from the valid goal space. Actually, the agent does not have access to such distribution during training, it is only used in the evaluation protocol. Equations and background notations have been reworked for better clarity.
>
>
> - **Experimental Results**
>
> Finally, the reviewer is not convinced by the experimental results and believes that FetchPickandPlace is the hardest environment we are addressing. We respectfully disagree.
> First, mazes like the ones used in our experiments are harder than they seem at a first glance, because the agent is blind to walls and cannot anticipate the many discontinuities of the goal space. Second, in terms of success coverage, mazes are very difficult tasks, as they require the policy to learn diverse behaviors that correspond to the many discontinuities in the goal space.  Our focus in this paper is on addressing the difficulties raised by such discontinuities. By contrast, more complex environments in terms of observation and action space such as FetchPickandPlace put all the stress on the RL algorithm, which is not our focus. Nevertheless, we see that SVGG still outperforms the baselines in AntMaze.

---

### Author Response · Authors · 2022-11-18
**General response to reviewers**

We warmly thank all reviewers for their feedback which helped us better appreciate some weaknesses of our work and provided a good basis for improving it.

Indeed, we clarified the positioning of the paper, better highlighting that SVGG brings the best of both worlds with respect to methods like MEGA which focuses more on exploration and GoalGAN which focuses more on successfully reaching known goals. This rewriting effort helped us better put forward the success coverage metrics that we use to show that focusing only on achieved goal coverage as MEGA does is not enough. We also improved the proof of the recovery property and the clarity in several places. All the modifications have been highlighted in red in the revisited version of the paper. Note that we also updated Fig. 1. The individual concerns of all reviewers are covered in dedicated responses.

---

### Decision · Program_Chairs · 2023-01-20

**Decision:**

Reject

**Justification For Why Not Higher Score:**

Weaknesses noted above.

**Justification For Why Not Lower Score:**

N/A

**Metareview: Summary, Strengths And Weaknesses:**

This paper proposes a Stein variational gradient descent approach to selecting goals for goal-conditioned RL agents to explore their environment without supervision, particularly for environments with hard exploration challenges. The objective for SVGD is to sample goals with probability based on: (1) is this goal within my prior distribution of goals? and (2) is this goal of intermediate difficulty for my current policy, based on my recent attempts to reach various goals?

Strengths:
- The high-level objective is interesting and the technical instantiation of it through SVGD is novel. The interpretation of the particles in SVGD as focusing the attention of the learner is interesting too.

Weaknesses:
- Empirical performance is not very convincing. As R2 points out, the results are somewhat different from our apriori expectations on when good goal sampling is most important, namely, the hardest environments like Fetch PickAndPlace --- in these envs, SVGG performance is rather weak. The authors argue that Fetch is mostly hard because of observation and action space rather than exploration-related challenges, but large observation and action spaces are also directly related to large exploration challenges.

- This work appears to mainly focus on achieving a more stable optimization of previously known desideratum, such as in GoalGAN. However, this framing is different from the current framing, and suggests a more incremental contribution. Further, alternative optimizers are not evaluated.

- Some sloppiness in exposition including errors in theory, noted during the response phase.

Final note:
There is some pretty extensive revision of the paper during the rebuttal period that appears to have improved the paper, but this demands quite extensive re-review, which is in my opinion a bit unfair to expect during the response phase.


Overall, the paper is more borderline than appears at first glance from the reviews, but after a careful reading, I have to fall on the side of recommending rejection for ICLR. However, this paper appears on track for better reception at the next venue, and I would encourage the authors to continue to improve and resubmit.

**Summary Of Ac-Reviewer Meeting:**

Clear weaknessses noted above.